# Structural insights into the clustering and activation of Tie2 receptor mediated by Tie2 agonistic antibody

Gyunghee Jo [1,2,5], Jeomil Bae [3,5], Ho Jeong Hong[2], Ah-reum Han[2], Do-Kyun Kim[2], Seon Pyo Hong[3], Jung A Kim[1], Sangkyu Lee [4], Gou Young Koh [1,3✉] & Ho Min Kim [1,2✉]

Angiopoietin (Angpt)-Tie receptor 2 (Tie2) plays key roles in vascular development and homeostasis as well as pathological vascular remodeling. Therefore, Tie2-agonistic antibody and engineered Angpt1 variants have been developed as potential therapeutics for ischemic and inflammatory vascular diseases. However, their underlying mechanisms for Tie2 clustering and activation remain elusive and the poor manufacturability and stability of Angpt1 variants limit their clinical application. Here, we develop a human Tie2-agonistic antibody (hTAAB), which targets the membrane proximal fibronectin type III domain of Tie2 distinct from the Angpt-binding site. Our Tie2/hTAAB complex structures reveal that hTAAB tethers the preformed Tie2 homodimers into polygonal assemblies through specific binding to Tie2 Fn3 domain. Notably, the polygonal Tie2 clustering induced by hTAAB is critical for Tie2 activation and are resistant to antagonism by Angpt2. Our results provide insight into the molecular mechanism of Tie2 clustering and activation mediated by hTAAB, and the structure-based humanization of hTAAB creates a potential clinical application.

[1] Graduate School of Medical Science and Engineering, Korea Advanced Institute of Science and Technology (KAIST), Daejeon 34141, Republic of Korea. [2] Center for Biomolecular & Cellular Structure, Institute for Basic Science (IBS), Daejeon 34126, Republic of Korea. [3] Center for Vascular Research, IBS, Daejeon 34141, Republic of Korea. [4] Center for Cognition and Sociality, IBS, Daejeon 34126, Republic of Korea. [5] These authors contributed equally: Gyunghee Jo, Jeomil Bae. ✉email: gykoh@kaist.ac.kr; hm_kim@kaist.ac.kr

Angiopoietin (Angpt)-Tie receptor (Tie1 and Tie2) signaling plays critical roles in morphogenesis and homeostasis of blood vessels and in vascular remodeling during vascular inflammation and tumor angiogenesis and metastasis[1–4]. Tie2 is a receptor tyrosine kinase (RTK) specifically expressed in vascular endothelial cells (ECs) and some hematopoietic stem cells and pericytes[1,3,5,6]. Tie2 possesses an extracellular domain (ECD) for ligand binding, a single-pass transmembrane domain, a cytoplasmic protein tyrosine kinase domain, and a C-terminal tail. The Tie2 ECD contains three immunoglobulins (Ig), three epidermal growth factors (EGF), and three fibronectin type III (Fn) domains (Fn1, Fn2, and Fn3) (Fig. 1a).

Angpt1 is an endogenous Tie2-agonistic ligand[7] consisting of an N-terminal super-clustering domain, a central coiled-coil domain, and a C-terminal fibrinogen-like domain (FLD) responsible for Tie2 binding[8]. The coiled-coil domain mediates Angpt1 dimerization or trimerization, and the N-terminal super-clustering domain is responsible for higher-order Angpt1 oligomerization. Thus, Angpt1 exists naturally as a heterogeneous mix of tetramers, pentamers, and predominantly higher-order oligomers, but a proper oligomeric status of Angpt1 is critical for Tie2

activation[9]. In contrast, Angpt2 primarily forms low-order oligomers, mainly dimers, and acts context dependently as a weak Tie2 agonist or antagonist, despite having ~60% amino acid sequence identity with Angpt1.

Crystal structures and biochemical analyses have demonstrated that the FLD of Angpt1 and Angpt2 binds to the same Ig2 domain of Tie2 with similar affinities. In addition, FLD binding does not cause any significant conformational rearrangements in the Tie2 ECD[10,11], indicating that Tie2 clustering and activation are most likely specified by Angpt oligomeric status. Recent structural studies, however, have shown that homotypic interactions between the membrane-proximal Fn domain of Tie2 mediate ligand-independent Tie2 dimerization, which also plays a crucial role in Angpt1-mediated Tie2 clustering and activation[12,13]. Nevertheless, the molecular mechanism of Tie2 clustering by native Angpt1 remains poorly understood, as do clustering by Angpt1 variants (pentameric COMP-Angpt1 and dimeric CMP-Angpt1)[14,15] and the reason for discrepancies in Angpt1 and Angpt2 activities.

Emerging evidence indicates that a Tie2 agonist could be a promising drug for several vascular diseases, including vascular

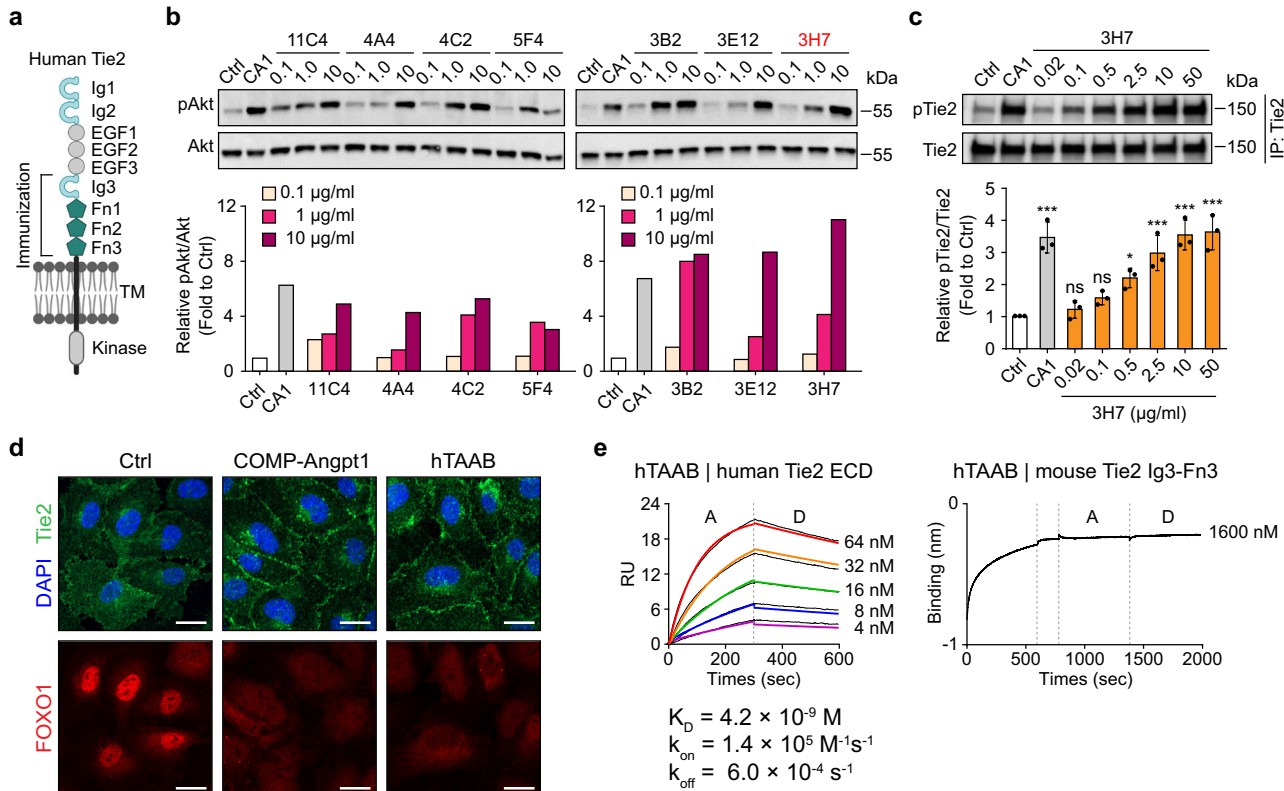

**Fig. 1 Generation of a Tie2-activating mouse monoclonal antibody. a** Schematic of the domain structure of the Tie2 receptor. Ig immunoglobulin-like domain, EGF epidermal growth factor-like domain, Fn fibronectin type III domain. **b** Immunoblot detection of Tie2 downstream signaling (Akt and p-Akt) in HUVECs stimulated with each of seven purified Tie2-activating antibodies (11C4, 4A4, 4C2, 5F4, 3B2, 3E12, and 3H7; 0.1, 1, and 10 μg/ml) for 30 min. The selected 3H7 clone is colored in red. No treatment for negative control (Ctrl) and COMP-Angpt1 (CA1, 1 μg/ml) for positive control. **c** Concentration-dependent Tie2 phosphorylation upon treatment of HUVECs with 3H7 (0.02, 0.1, 0.5, 2.5, 10, 50 μg/ml). Immunoblot (top) and densitometric analyses (bottom) of p-Tie2/Tie2 ratios are shown. Data from three independent experiments ($n = 3$) were analyzed and expressed as mean ± SD (*$P = 0.0104$, ***$P < 0.001$ vs. control). $P$ values by one-way ANOVA test followed by Dunnett's multiple comparisons test. ns, not significant. **d** Representative confocal images of HUVECs showing hTAAB-induced Tie2 translocation to cell–cell contacts (top) and hTAAB-induced nuclear clearance of FOXO1 (bottom). Serum-starved HUVECs were treated with hTAAB (10 μg/ml) or COMP-Angpt1 (1 μg/ml) for 30 min. Goat anti-human Tie2 and Alexa Fluor 488-conjugated donkey anti-goat antibodies were used for Tie2 visualization, and rabbit anti-FOXO1 and Alexa Fluor 594-conjugated donkey anti-rabbit antibodies were used for FOXO1 visualization. Scale bars, 20 μm. DAPI, 4′, 6-diamidino-2-phenylindole. Similar results were observed in three independent experiments. **e** Binding kinetics of hTAAB for human (left) or mouse (right) Tie2 determined by SPR and BLI analysis, respectively. The equilibrium dissociation constant ($K_D$, M) was calculated as the ratio of off-rate to on-rate ($k_{off}/k_{on}$). Kinetic parameters were determined with the global fitting function of Biacore Insight Evaluation Software using a 1:1-binding model.

retinopathies, sepsis, glaucoma, and cancers, by enhancing vascular stability, normalization, and rejuvenation while reducing vascular leakage and inflammation[3,16–19]. Native Angpt1 is difficult to handle because of its poor solubility and heterogeneity, and most previous efforts to develop Tie2 agonists have focused on Angpt FLD-based protein engineering, e.g., COMP-Angpt1, CMP-Angpt1, Hepta-Angpt1, and FLD-scaffold nanoparticles[14,15,20,21]. However, limitations that must be overcome for clinical application include low-yield production, short half-life, non-specific binding to most tissues, and potential immunogenicity.

Tie2-activating antibodies also have been developed as a useful alternative Tie2 agonist[22,23], but their binding mode to Tie2 and role in Tie2 activation remain elusive. A recently developed humanized monoclonal anti-Angpt2 antibody, ABTAA (Ang2-binding and Tie2-activating antibody), activates Tie2 and stimulates its downstream signaling by inducing oligomerization of Angpt2 bound to Tie2[16,19]. However, ABTAA functions only in the presence of Angpt2, so that its therapeutic efficacy largely depends on Angpt2 levels in a pathological microenvironment, which is difficult to control. Moreover, epitopes in Angpt2 for ABTAA binding and the oligomerization status of active signaling complexes (ABTAA/Angpt2/Tie2 complex) remain unclear. The optimal dose of ABTAA for reducing unanticipated aggregations between ABTAA and Angpt2 also needs to be carefully explored for clinical application.

Here, we develop a Tie2-activating antibody that directly targets the membrane-proximal Fn3 domain of Tie2 and elicits biological responses similar to those that natural ligands trigger. The crystal structure of this antibody in complex with Tie2 reveals the antigen's epitope and the antibody's paratope. In addition, negative-stain EM analysis and functional characterization offer mechanistic insights into how our Tie2-activating antibody can induce polygonal Tie2 clustering for Tie2 activation. Successful structure-based humanization of this mouse Tie2-activating monoclonal antibody opens new avenues for future clinical applications.

## Results

**Generation of Tie2-activating mouse monoclonal antibody.** The ECD of Tie2 forms dimers via its membrane-proximal Fn3, regardless of ligand binding[12,13]. Multimeric Angpt1 binding to ligand-binding domains (LBDs) further crosslinks these preformed Tie2 dimers into higher-order oligomers, or Tie2 clusters for Tie2 activation and downstream signaling. However, the LBDs of preformed Tie2 homodimers are too far apart (~260 Å)[12] for a single antibody targeting LBD to induce Tie2 clustering. Based on these notions, we postulated that an anti-Tie2 antibody that binds to the membrane-proximal Tie2 Fn domain might induce Tie2 oligomerization and activation as multimeric Angpt1 does.

To this end, we immunized BALB/c mice with a recombinant protein encompassing Ig3–Fn3 of human Tie2 (hTie2) twice weekly for 6 weeks and then fused B lymphocytes isolated from the spleens with cultured myeloma cells (SP2/0) (Fig. 1a). After selection with HAT (hypoxanthine-aminopterin-thymidine) medium, we performed subsequent clonal expansion and ELISA-based Tie2-binding assays with culture media, yielding 37 positive hybridoma clones (Supplementary Fig. 1a). After discarding two clones because of insufficient antibody production, we screened the remaining 35 for a potential candidate by monitoring the level of Akt phosphorylation in primary cultured human umbilical vein endothelial cells (HUVECs) (Supplementary Fig. 1b). Of the candidate clones that emerged (11C4, 4A4, 4C2, 5F4, 3B2, 3E12, and 3H7), we chose 3H7 as the best option because of its adequate and efficient growth and expansion and antibody production. In addition, the antibody produced from

the 3H7 clone induced the most potent Akt and Tie2 phosphorylation in a dose-dependent manner (Fig. 1b, c). We designated the antibody produced by the 3H7 clone as a human Tie2-activating antibody (hTAAB).

Treatment of HUVECs with hTAAB-induced Tie2 translocation to cell–cell contact sites as well as FOXO1 translocation from the nucleus to the cytosol, similar to treatment with the Tie2 agonist COMP-Angpt1 (Fig. 1d). Binding kinetic analyses using surface plasmon resonance (SPR) and biolayer interferometry revealed that hTAAB binds to the ECD of hTie2 with nanomolar affinity ($k_{on} = 1.4 \times 10^5$ M$^{-1}$ s$^{-1}$; $k_{off} = 6.0 \times 10^{-4}$ s$^{-1}$; $K_D = 4.2 \times 10^{-9}$ M), without exhibiting any binding to mouse Tie2 (mTie2) Ig3–Fn3 (Fig. 1e). PCR amplification with the cDNA of the hybridoma 3H7 clone using a mouse Ig-Primer followed by DNA sequencing revealed complete nucleotide sequences encoding the hTAAB heavy- and light-chain variable regions. The amino acid sequence of the heavy and light chains of hTAAB disclosed 90.8% sequence homology to the mouse germline heavy-chain variable gene *mIGHV1-61\*01* and 98.9% sequence homology to the light-chain gene *mIGKV9-120\*01*. Taken together, we generated hTAAB as a hTie2-activating mouse monoclonal antibody that targets Ig3–Fn3 of hTie2 but not of mTie2.

**The crystal structure reveals two hTAAB Fabs bind to a V-shaped Tie2 dimer.** To delineate the epitope for hTAAB binding, we identified the minimal domain of Tie2 for hTAAB binding. Because hTie2 Ig3–Fn3 was used to immunize mice for antibody generation, we first produced three different recombinant proteins containing Ig3–Fn3, Fn1–3, or Fn2–3 of hTie2 (Fig. 1a, see the architecture of Tie2). Using size-exclusion chromatography (SEC), we tested their binding to recombinant chimeric hTAAB Fab, which was constructed from fusion of the hTAAB heavy-chain variable region to the human γ1 constant region and of the light-chain variable region to the human κ constant region. Chromatography profiles of chimeric hTAAB Fab alone showed a single peak that shifted forward when chimeric hTAAB Fab was mixed with hTie2 Ig3–Fn3, Fn1–3, or Fn2–3 (Supplementary Fig. 2a). Moreover, chimeric hTAAB Fab and these hTie2 domains co-eluted. These results indicate that hTie2 Fn2–3 is sufficient for hTAAB binding and that hTie2 Fn2–3 and chimeric hTAAB Fab can form a stable complex (Supplementary Fig. 2a, yellow).

Next, we determined the crystal structure of the hTie2 Fn2–3/chimeric hTAAB Fab complex at 2.1 Å resolution (Fig. 2a and Table 1) by molecular replacement using durvalumab Fab (PDB: 5X8M) and Tie2 Fn2–3 structures (PDB: 5MYA chain B) as search models[12,24]. The crystallographic asymmetric unit contained a 2-fold symmetrical, hetero-tetrameric hTie2 Fn2–3/chimeric hTAAB Fab complex (2:2 stoichiometry). Two chimeric hTAAB Fabs were bound to the lateral side of each hTie2 Fn3 domain at about a 15° tilt with respect to the plane perpendicular to the 2-fold axis (Fig. 2a, right). This crystal structure unveiled two major epitopes of the Fn3 domain for hTAAB binding, which are indicated in dark orange for heavy-chain binding and light orange for light-chain binding (Fig. 2b, c, top). Two hTie2 Fn2–3 monomers in the hetero-tetrameric hTie2 Fn2–3/chimeric hTAAB Fab complex were found to share the same overall structure and to be similar to previously reported crystal structures of Tie2 Fn2–3 (PDB: 5MYB; red) and Fn1–3 (PDB: 5UTK; green)[12,13], indicating a rigid linkage between Fn2 and Fn3 (Supplementary Fig. 2b; the Cα root mean squared deviation was ~1.050 Å between each hTie2 Fn2–3 monomer, 1.506 Å with 5MYB and 1.999 Å with 5UTK). Of note, the configuration and interface between hTie2 dimers in the hTie2 Fn2–3/chimeric hTAAB Fab complex were also consistent with those found in

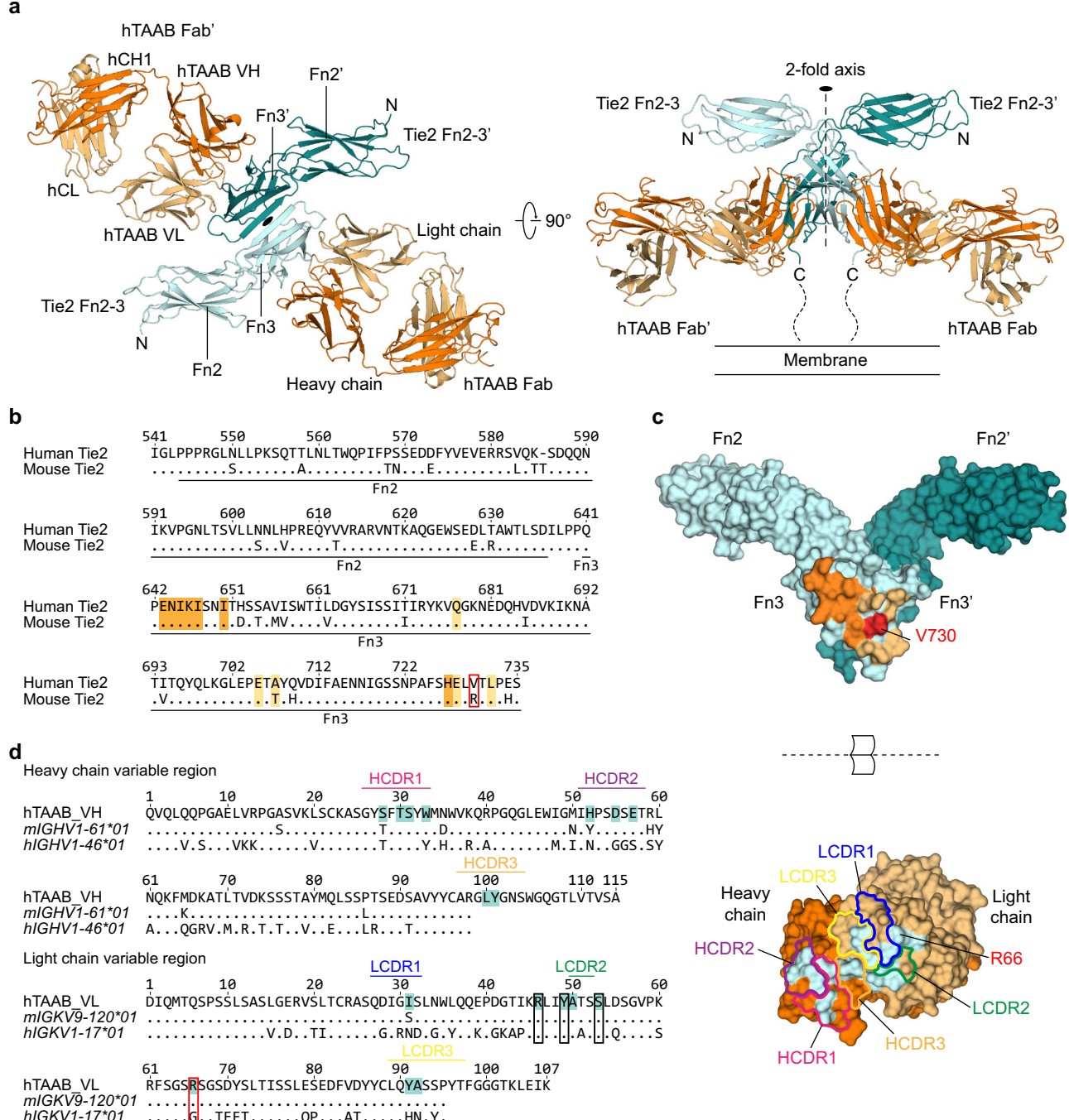

**Fig. 2 Overall structure of hTAAB Fab in complex with Tie2 Fn2-3. a** Overall structure of the 2:2 hTie2 Fn2–3/chimeric hTAAB Fab complex. The 2-fold axis of the dimeric complex is indicated as a black ellipse. The heavy and light chains of chimeric hTAAB Fab are colored in dark and light orange, respectively. The hTie2 Fn2–3 dimer is colored in teal and cyan. **b** Sequence comparison of human and mouse Tie2 Fn2–3 domains. The dots in the mouse Tie2 sequence indicate residues that are identical to human Tie2. **c** The surface representations are open-book views showing the interacting interfaces of Tie2 Fn2–3 (top) and chimeric hTAAB Fab (bottom). The color scheme of hTie2 Fn2–3 and chimeric hTAAB Fab is identical to that in (**a**), but inverted for their respective binding surface. Six CDR loops of chimeric hTAAB Fab are indicated with different color lines (bottom). **d** Sequence comparison of the variable region of hTAAB heavy (top) and light (bottom) chains with closest human and mouse variable region germline genes. The dots in mouse and human germline sequences indicate the residues that are identical to hTAAB. **b–d** hTie2 residues that interact with the heavy and light chain of chimeric hTAAB Fab are colored dark and light orange, respectively, and V730, which interacts with both heavy and light chains, is highlighted in red (**b**, **c**, top). The chimeric hTAAB Fab residues that interact with hTie2 Fn3 are colored cyan (**c**, bottom, and **d**).

previous studies (5MYB and Dimer2 model of 5UTK)[12,13] (Supplementary Fig. 2c, d).

These results indicate that hTAAB binding triggered no conformational changes in the hTie2 homodimer. Four domains of chimeric hTAAB Fab—variable heavy domain (VH), constant heavy domain (CH), variable light domain (VL), and constant light domain (CL)—adopted a typical Ig domain fold consisting of a pair of β-sheets. All complementarity-determining regions (CDRs) of the heavy chain (HCDR1, HCDR2, and HCDR3) and light chain (LCDR1, LCDR2, and LCDR3) participated in contact

**Table 1 Data collection and refinement statistics.**

| | hTie2 Fn2–3/chimeric hTAAB Fab complex |
|---|---|
| *Data collection* | |
| Wavelength (Å) | 0.97940 |
| Space group | P1 |
| Cell dimensions | |
| *a, b, c* (Å) | 45.08, 77.34, 121.85 |
| α, β, γ (°) | 103.82, 96.89, 90.59 |
| Resolution (Å) | 44.72-2.09 (2.169-2.09) |
| $R_{merge}$ (%)[a] | 9.9 (50.9) |
| $I / \sigma I$ | 7.126 (1.217) |
| Completeness (%) | 91.04 (88.61) |
| Redundancy | 3.6 (3.4) |
| *Refinement* | |
| Resolution (Å) | 2.09 |
| No. of reflections | 84,929 |
| $R_{work}/R_{free}$ (%)[b] | 18.3/23.3 |
| No. of atoms | |
| Protein | 9621 |
| Water | 737 |
| Average *B*-factors | 39.0 |
| R.m.s. deviations | |
| Bond lengths (Å) | 0.0073 |
| Bond angles (°) | 0.877 |
| Ramachandran | |
| Favored (%) | 95.9 |
| Outlier (%) | 0.08 |
| PDB ID code | 7E72 |

Values in parentheses are for highest-resolution shell.
[a]$R_{merge} = \sum_{hkl}\sum_i |I_{hkli} - \langle I_{hkli}\rangle| / \sum_{hkl}\sum_i \langle I_{hkli} \rangle$.
[b]$R_{cryst} = \sum_{hkl} ||F_o| - |F_c|| / \sum |F_o|$.

with the hTie2 Fn3 domain (Fig. 2c, bottom). In addition to six CDRs, residues R46, Y49, S53, and R66 of light-chain framework regions (FRs) also interacted with the hTie2 Fn3 domain (Fig. 2d). Collectively, the findings regarding the crystal structure revealed that two hTAAB Fabs bind to the lateral sides of the V-shaped Tie2 dimer, specifically to the Fn3 domain.

**Interactions of Tie2 Fn3 with hTAAB and homotypic Fn3 interactions**. Chimeric hTAAB Fab binds to hTie2 Fn3 through a large interaction interface with a total buried surface area of 897.5 Å$^2$, consisting of ~57% heavy-chain contacts (512.8 Å$^2$; dark orange) and 43% light-chain contacts (384.7 Å$^2$; light orange) (Fig. 2c). The binding interfaces between chimeric hTAAB Fab and hTie2 Fn3 can be divided into three distinct regions, the A, B, and C regions (Fig. 3a, b and Supplementary Fig. 3).

A-region interactions are mainly mediated by ionic interactions and hydrogen bonds between the hTAAB heavy chain (HCDR1 and HCDR2) and the hTie2 Fn3 domain. Residues on hTAAB HCDR1 (S28, T30, S31, and W33) and HCDR2 (H52, D55, and E57) form multiple interactions with residues on hTie2 Fn3 βA (E643, N644, I645 (main chain), K646, I647 (main chain)) and βG (H727) (Fig. 3a, top left and bottom left, and 3c). In the B-region, HCDR3, LCDR1, LCDR2, and LCDR3 of hTAAB cover a wide surface of hTie2 Fn3 through hydrophobic interactions, with a total buried surface area of 413.5 Å$^2$. Residues I647, I650, A707, V730, and L732 of hTie2 Fn3 form a network of hydrophobic core interactions with residues L100 and Y101 on HCDR3, and I31, Y49, A50, Y91, and A92 on LCDRs (Fig. 3a, bottom left, and 3d). These hydrophobic interactions are stabilized by neighboring C-region interactions, which include hydrogen bonds and electrostatic interactions between hTie2 Fn3 residues (Q677, E705, and E728) and hTAAB light-chain residues

(S53, R66, and R46), respectively (Fig. 3a, bottom center, and 3c). Of interest, residues of hTAAB light chains involved in C-region interactions are located on the β-strand of FRs (R46 on βC', S53 on βC", and R66 on βD) rather than the CDR loop. Although most hTie2 Fn3 residues involved in hTAAB interactions are highly conserved among various species (Supplementary Fig. 3a), the hydrophobic valine (V730) of hTie2 at the center of the hTAAB binding interface is substituted with a long, charged side chain (arginine) in mouse and rat (Fig. 2b and Supplementary Fig. 3a), explaining why hTAAB cannot bind mouse Tie2 (Fig. 1e, right). However, all epitopes of hTAAB are conserved between human and monkey Tie2, suggesting that hTAAB would have cross-reactivity to monkey Tie2 (Supplementary Fig. 3a). In addition, hTAAB is unlikely to bind Tie1 because of the low sequence similarity between the Fn2–3 domains of Tie2 and Tie1, particularly the residues involved in hTAAB interactions (Supplementary Fig. 3b).

Consistent with previous Tie2 apo-structures (5MYB and Dimer2 model of 5UTK)[12,13], the Fn3–Fn3' interface is composed of hydrogen bonds between the main-chain atoms of homotypic Fn3 βC' (D682~K690), which form a continuous antiparallel β-sheet (Fig. 3a, top right). This dimeric interface is further stabilized by hydrophobic interactions involving V685 and V687, as well as by reciprocal electrostatic interactions, specifically D682-N691', D682'-N691, Y697-Q683', Y697'-Q683, K700-E703', and K700'-E703 (Fig. 3a, bottom right). The wide buried interface (704 Å$^2$) between Fn3–Fn3', their low solvation free energy gain of -4.7 kcal/mol (Pisa server analysis), and the consistent V-shaped configuration of Tie2 homodimer regardless of the presence of hTAAB imply that specific interactions between homotypic membrane-proximal Fn3 domains contribute to ligand-independent Tie2 homodimer formation.

**Binding of hTAAB IgG induces polygonal assembly of Tie2 dimers**. Multimeric Angpt1 or COMP-Angpt1 can induce higher-order clustering of Tie2, which is key for activating Tie2 and its downstream signaling in ECs for vascular stabilization[9,14]. Intriguingly, hTAAB IgG1, but not hTAAB Fab alone, induced concentration-dependent activation of Tie2 and Akt in primary cultured HUVECs (Fig. 4a, b). This result implied that the binding of hTAAB Fab to Tie2 itself is insufficient to induce Tie2 clustering and activation. In other words, hTAAB IgG1 may facilitate Tie2 clustering via its bivalent Fab arms. However, we found that the distance between the two C-termini of Fab heavy chains in the hetero-tetrameric hTie2 Fn2–3/chimeric hTAAB Fab complex is greater than 170 Å (Fig. 2a), suggesting that the two Fabs that bind to a homotypic Tie2 dimer are derived from two different IgG molecules. Unlike the crystal structure of the hetero-tetrameric hTie2 Fn2–3/chimeric hTAAB Fab complex, SEC-multiangle light-scattering (MALS) analysis revealed that hTie2 Fn2–3 and hTie2 ECD were monomeric in solution. Moreover, hTAAB binding to hTie2 Fn2–3 or hTie2 ECD could not induce their oligomerization (Fig. 4c). Our data and previous reports[12,13] were contradictory regarding the multimeric status of recombinant Tie2 ectodomain in solution. However, ligand-independent Tie2 dimerization can occur in a cellular membrane under physiological conditions through spatial constraints imposed by full-length Tie2 (Fig. 5d). This dimerization particularly can occur through the YIA sequence between the catalytic and activation loops in the intracellular kinase domain[25], together with the homotypic Fn3–Fn3' interactions shown in the crystal structure.

To mimic ligand-independent Tie2 dimers in a physiological cellular membrane, we artificially generated a constitutive Tie2 dimer. We did so by mutating two residues, D682 and N691,

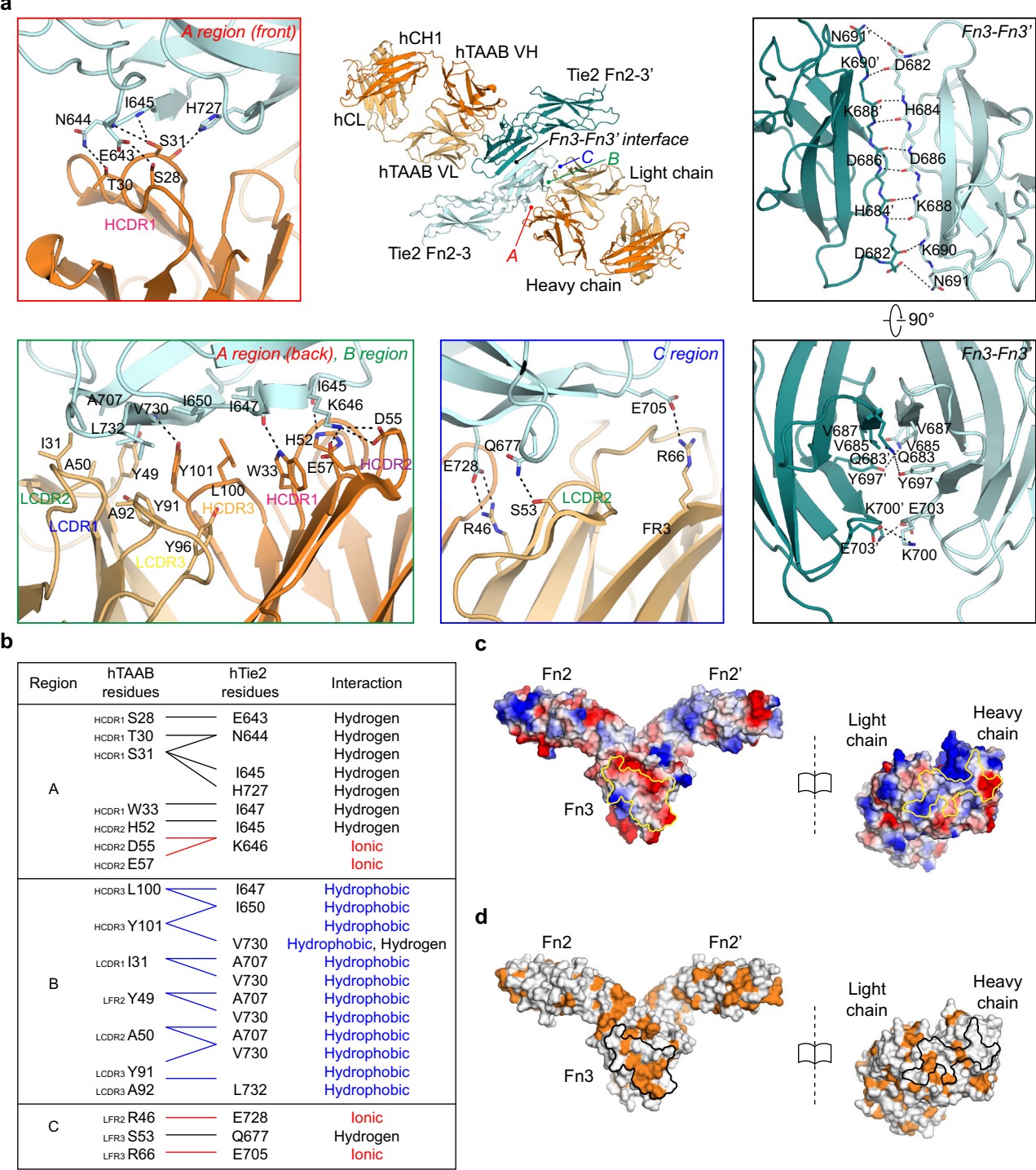

**Fig. 3 Interactions of Tie2 Fn3 with hTAAB and homotypic Fn3 interactions. a** Close-up view of key molecular interactions (A, B and C regions). Residues involved in the hTie2 Fn3/chimeric hTAAB Fab interaction (red, green, and blue boxes) and hTie2 dimerization (black box) are displayed as sticks and labeled. Hydrogen bonds and electrostatic interactions are indicated by dashed lines. **b** Key amino acid residues of A, B, and C regions involved in hTAAB binding. Interaction residues of hTAAB are listed on the left, and the corresponding interaction residues of Tie2 are listed on the right. Interactions between residue partners are indicated in red (ionic interaction), black (hydrogen bond), and blue (hydrophobic interaction). **c** Electrostatic potential of the hTie2 Fn2–3/chimeric hTAAB Fab complex, calculated according to the Poisson–Boltzmann equation in PyMOL. The structures are shown as open-book views and surface representations, with color maps reflecting electrostatic properties (blue, positively charged; red, negatively charged). The interacting regions are highlighted by a yellow line. **d** Hydrophobic residues in hTie2 Fn2-3 and chimeric hTAAB Fab are presented on the surface, and the interaction interface is marked with a black line.

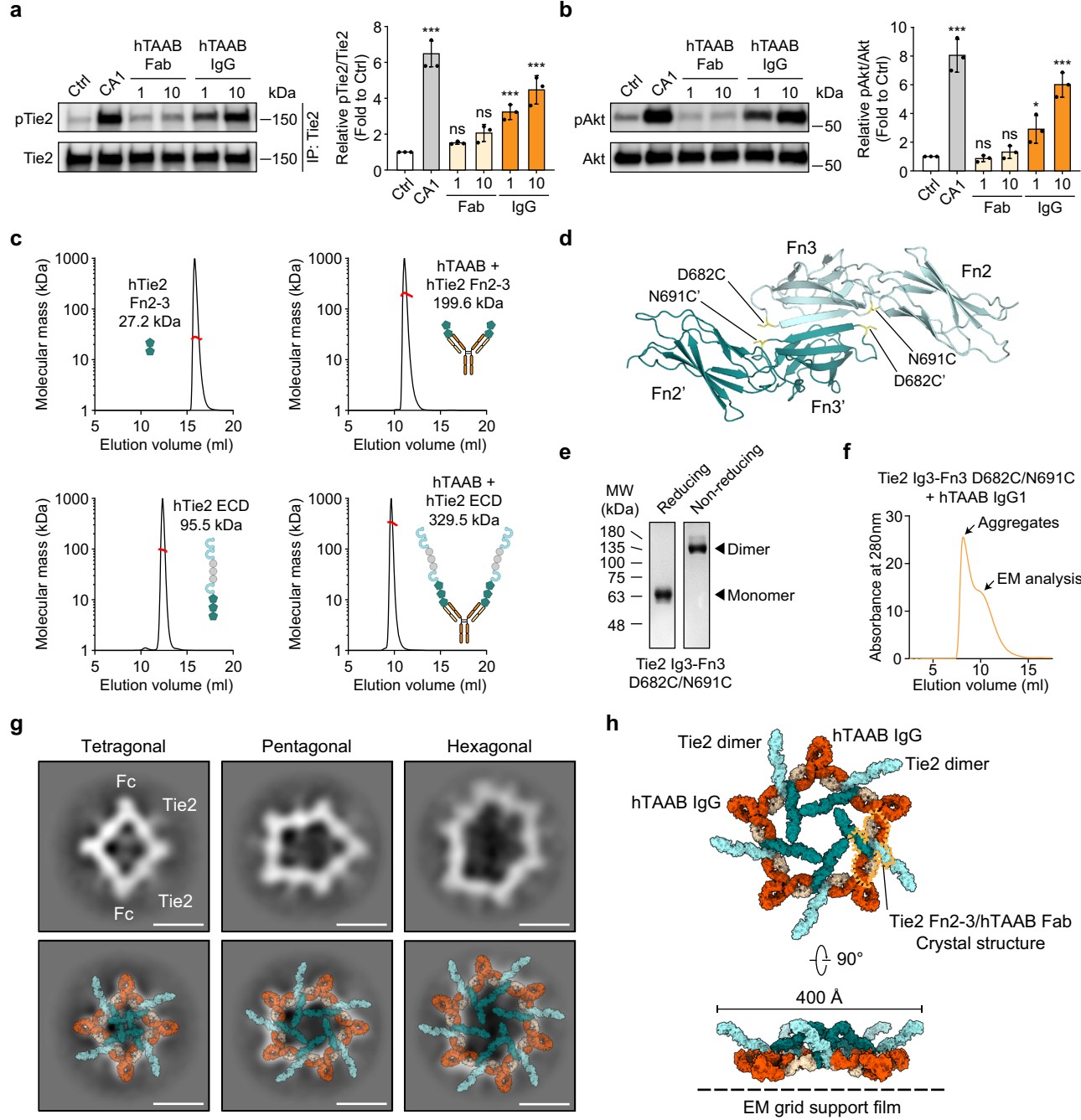

which are involved in homotypic Fn3–Fn3' interactions, to cysteines (D682C and N691C) to introduce a disulfide bond (Fig. 4d, e). We then incubated the purified hTie2 Ig3–Fn3 D682C/N691C dimer with hTAAB IgG1 at a 1:1 ratio to investigate how full-length hTAAB IgG1 induces clustering of the preformed Tie2 dimer. After removal of aggregates by SEC, we examined the structural arrangement of the resulting complexes (peak of the shoulder) using negative-stain electron microscopy (EM) (Fig. 4f and Supplementary Fig. 4). Surprisingly, 2D classification of negative-stain EM particles showed that full-length hTAAB IgG1 cross-linked hTie2 Ig3–Fn3 D682C/N691C dimers into tetragonal, pentagonal, and hexagonal higher-order assemblies (Fig. 4g). These polygonal assemblies featured an Fc domain of IgG1 at each vertex and one Tie2 dimer bridging two IgG1s by binding to one of their Fab arms. Although the flexibility of loop between Ig3 and Fn1 domain made Ig3 domain

appear diffuse in 2D averages of negative-stain EM, the rigid Fn1–3 domain of hTie2 Ig3–Fn3 D682C/N691C dimer crossing the side of the polygon was distinguishable. Based on these observations from negative-stain EM and our crystal structure of the hTie2 Fn2–3/chimeric hTAAB Fab complex, we could generate a 3D model of pentagonal assemblies (5-to-5 hTAAB IgGs and hTie2 Ig3–Fn3 D682C/N691C dimers) (Fig. 4h).

**Ligand-independent Tie2 dimerization is essential for hTAAB-mediated Tie2 clustering and activation.** Next, we assessed the biological relevance and significance of these Tie2/hTAAB polygonal assemblies in the cell membrane. For this purpose, we used the constitutive Tie2 dimer (D682C/N691C) and created a constitutive Tie2 monomer (V685D/V687D/K700E) by disrupting the Fn3–Fn3' dimeric interface (Fig. 5a) with a full-length Tie2-GFP construct. Then, we used these constructs to transiently

**Fig. 4 hTAAB IgG binding mediates polygonal assembly of Tie2 dimers. a** Immunoblot analysis for relative phosphorylation ratios of Tie2 after treatment of HUVECs with chimeric hTAAB Fab or hTAAB mouse IgG1 (1 and 10 µg/ml) (left). Densitometric analyses of p-Tie2/Tie2 ratios are shown (right). Data from three independent experiments ($n = 3$) were analyzed and expressed as mean ± SD (***$P < 0.001$ vs. control). **b** Immunoblot analysis for relative phosphorylation ratios of Akt after treatment of HUVECs with chimeric hTAAB Fab or hTAAB IgG1 (1 and 10 µg/ml) (left). Densitometric analyses of the p-Akt/Akt ratios are shown (right). Data from three independent experiments ($n = 3$) were analyzed and expressed as mean ± SD (*$P = 0.0304$, ***$P < 0.001$ vs. control). **a**, **b** $P$ values by one-way ANOVA test followed by Dunnett's multiple comparisons test. ns, not significant. **c** Size-exclusion chromatography-multiangle light-scattering (SEC-MALS) analysis of hTie2 Fn2–3, hTie2 ECD, hTAAB/hTie2 Fn2–3 complex, and hTAAB/hTie2 ECD complex. Samples were run on a Superdex 200 Increase 10/300 GL column in PBS at 0.5 ml/min, at protein concentrations of 3 mg/ml (hTie2 Fn2–3) or 1 mg/ml (all others). The molecular mass (kDa) and absorbance at 280 nm are plotted against elution volume (ml). **d** Design of the Tie2 dimeric mutant. Residues D682 and N691 (yellow) on both ends of the antiparallel β-sheet interaction interface between Fn3 domains were mutated to cysteine to allow the formation of two disulfide bonds, thus generating a constitutive Tie2 Ig3–Fn3 dimer. **e** Purified hTie2 Ig3–Fn3 D682C/N691C was analyzed by SDS-PAGE and Coomassie blue staining under reducing and non-reducing conditions. Similar results were observed in three independent experiments. **f** Size-exclusion chromatography of hTie2 Ig3–Fn3 D682C/N691C in complex with hTAAB IgG1. Purified hTie2 Ig3–Fn3 D682C/N691C was incubated with hTAAB IgG1 for 2 h at a molar ratio of 2:1 (Tie2 monomer:hTAAB IgG1) and applied to size-exclusion chromatography. The fraction used for negative-stain EM and 2D class average is indicated as EM analysis. **g** Representative 2D class averages of Tie2 dimeric mutant/hTAAB IgG1 complex (Top). Cyclical higher-order structure of the Tie2 dimer/hTAAB IgG1 complex in 4-to-4, 5-to-5, and 6-to-6 assemblies were modeled based on the 2D class averages of tetragonal, pentagonal, and hexagonal closed-ring structures, respectively, using crystal structures of the Tie2 Fn2–3/chimeric hTAAB Fab complex, Tie2 Ig1–Fn1 (PDB: 4K0V[11]) and Fc fragment of human IgG1 (PDB: 5VGP[57]) (bottom). Scale bars, 20 nm. **h** 3D Model of the polygonal 5-to-5 assembly of the Tie2 dimer/hTAAB IgG1 complex observed on the negative-stain EM grid. The 2:2 Tie2 Fn2–3/chimeric hTAAB Fab complex in an asymmetric unit of the crystal is indicated by a red line.

transfect HEK293T cells, which do not endogenously express Tie2. After confirming proper plasma membrane expression of full-length Tie2-GFP wild-type (WT) as well as constitutive dimeric and monomeric Tie2 mutants in these cells, we monitored the clustering of GFP-tagged Tie2 WT and mutants. In live-cell imaging, cluster of WT Tie2 and constitutive Tie2 dimers are formed at the cell surface after the addition of hTAAB or COMP-Angpt1, whereas constitutive Tie2 monomers failed to form clusters (Fig. 5b, c). In fluorescence size-exclusion chromatography (FSEC) with detergent-solubilized full-length Tie2-GFP (Tie2 WT, dimeric and monomeric Tie2 mutant), Tie2 WT and dimeric Tie2 mutant eluted at similar volumes, whereas monomeric Tie2 mutant eluted later, indicating that Tie2 is a dimer in the plasma membrane (Fig. 5d).

We next examined the effect of hTAAB IgG1 or COMP-Angpt1 on Tie2 activation by monitoring Tie2 and Akt phosphorylation under the same conditions. Disruption of the Fn3–Fn3' dimeric interface (constitutive Tie2 monomers) abolished hTAAB- and COMP-Angpt1-induced Tie2 and Akt phosphorylation, whereas WT Tie2 and constitutive dimeric Tie2 were similarly activated by these two Tie2 agonists (Fig. 5e, f). These results indicate that WT Tie2 exists as a homotypic dimer in the plasma membrane and that ligand-independent homotypic Tie2 dimerization, which is attributable to intermolecular interactions between Fn3 domains, is essential for hTAAB-mediated Tie2 clustering and activation (Fig. 5g).

**Humanized TAAB activates Tie2 and its downstream signaling in HUVECs.** Potent Tie2-agonistic activity of hTAAB holds promise in clinical applications for vascular diseases. Therefore, we sought to humanize mouse hTAAB while maintaining its capacity to bind and activate Tie2 but minimizing its immunogenicity, which is attributable to its murine origin. Because of variation in length, sequence, and the number of disulfide bonds at the hinge between the Fab and Fc domains, the hinge flexibility of IgG subclasses is slightly variable, as are the Fab–Fab angle and Fab orientation relative to the Fc domain for each subclass[26]. Given that the Fab–Fab angle and Fab orientation of hTAAB may be key determinants of polygonal assembly of Tie2, we investigated whether this hinge flexibility influences Tie2 activation. To this end, we produced chimeric IgG1, IgG2, or IgG4 containing the variable region of hTAAB (Supplementary Fig. 5a) and compared their Tie2-agonistic activities with those of parental mouse hTAAB. Tie2

and Akt phosphorylation were correlated with the hinge flexibility of the IgG subclass (IgG1 > IgG4 > IgG2). The IgG2 isotype, with the most rigid hinge, showed the lowest Tie2-agonistic activity (Fig. 6a, b).

Although the majority of Tie2 clusters induced by hTAAB were a polygonal shape, a few linear Tie2 clusters were observed (Supplementary Fig. 4a, red box). Therefore, to investigate whether the different Tie2-agonistic activities of IgG subclasses are related to the Tie2-clustering mode (polygonal vs. linear), we performed a shape-based statistical analysis of the heterogeneous Tie2 Ig3–Fn3 dimer/hTAAB IgG (human chimeric IgG1, IgG2, or IgG4) complexes using negative-stain EM. We classified individual particles into six categories according to their shape (linear, <tetragonal, tetragonal, pentagonal, hexagonal, and >hexagonal) and compared the Tie2-activation potency among IgG subclasses (Supplementary Fig. 5b, c). Surprisingly, we found that the IgG2 subclass (with the lowest Tie2-agonistic activity) mostly induced linear Tie2 clustering, whereas various types of polygonal assemblies were observed in the IgG1 and IgG4 subclasses, which exhibited higher Tie2-agonistic activity than IgG2 (tetragonal + pentagonal + hexagonal + >hexagonal clustering: 73.1%, 11.3%, and 69.5% for IgG1, IgG2, and IgG4, respectively) (Fig. 6c). The proportion of linear cluster correlated with the hinge rigidity of the IgG subclass (IgG2 > IgG4 > IgG1) and inversely correlated with Tie2 and Akt phosphorylation (IgG1 > IgG4 > IgG2). These results indicate that polygonal Tie2 clustering induced by hTAAB is critical for Tie2 activation. Based on these observations, we chose the IgG1 format for hTAAB humanization.

Next, using the IMGT/Domain Gap Align tool[27] from the International ImmunoGeneTics information system (IMGT) (http://www.imgt.org), we defined CDR boundaries and selected human germline genes *IGHV1-46*01* and *IGKV1-17*01* as human FR donors because they exhibit the highest sequence identities to variable heavy-chain (66%) and light-chain (68%) regions of mouse hTAAB (Fig. 6d and Supplementary Fig. 3c, d). We then performed conventional CDR grafting by simply replacing CDR portions in the selected human germline FR donors with corresponding CDRs of hybridoma mouse antibody hTAAB. To maintain the conformation of hTAAB HCDR2, we also replaced S59 with a R59 residue flanking the HCDR2 of hTAAB heavy chains in the humanized antibody. The CDR-grafted variable regions of humanized Tie2-activating antibody (hzTAAB) heavy and light chains (hzTAAB-H1 and hzTAAB-L1) were cloned into a pOptiVEC-TOPO vector

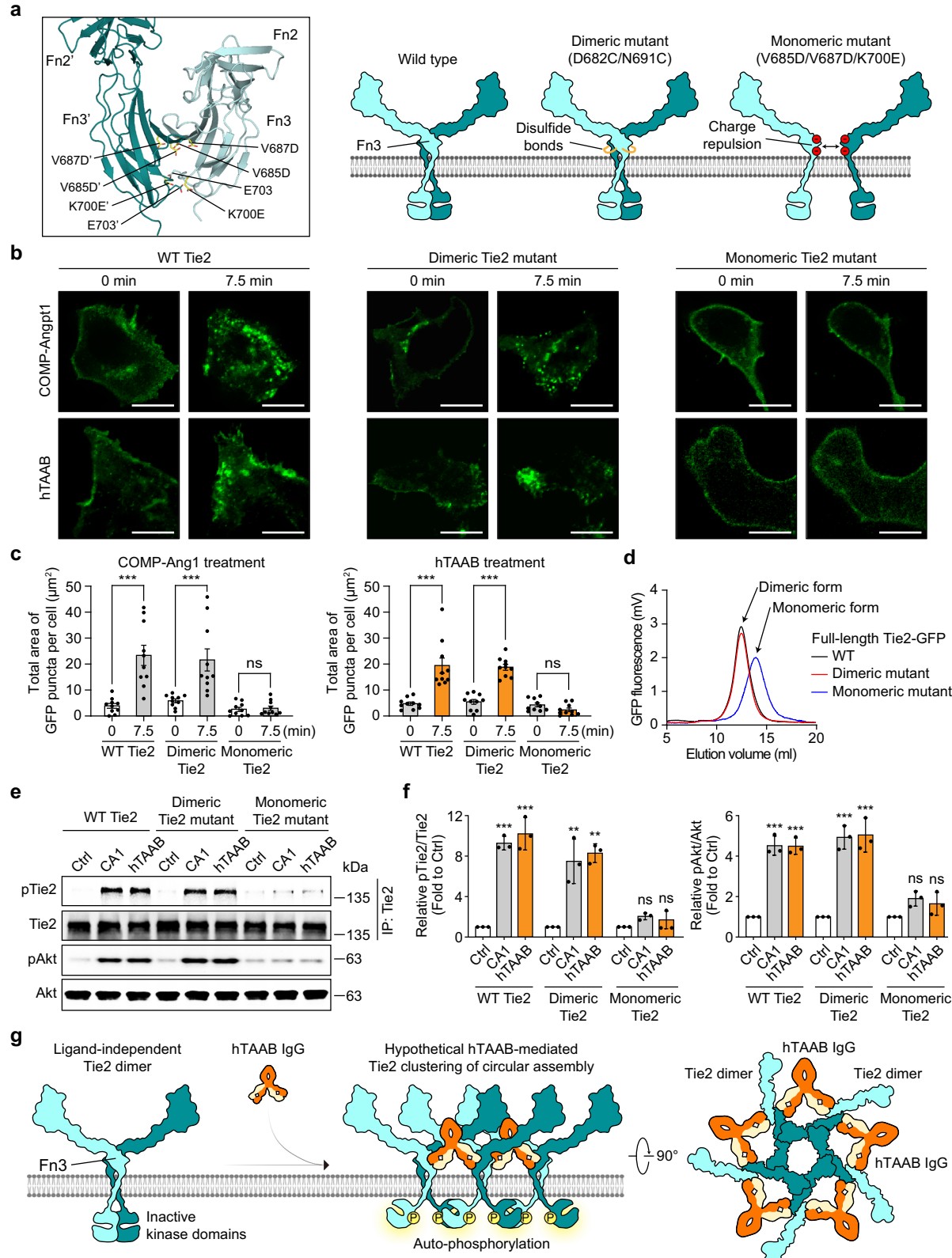

containing the constant region of human Ig γ1 heavy chain (GenBank accession number: AWK57454.1) and a pcDNA3.3-TOPO vector containing the constant region of human Ig κ light chain (GenBank accession number: AAA58989.1), respectively. After co-expressing hzTAAB-H1 and hzTAAB-L1 in HEK293F cells and purifying the antibodies to homogeneity as full-length IgG1 by protein A affinity chromatography and SEC (Supplementary

Fig. 5d, e), we evaluated the binding kinetics of hzTAAB-H1L1 to hTie2 ECD using SPR. hzTAAB-H1L1 bound to hTie2 with a much lower affinity and faster dissociation rate ($k_{on} = 2.5 \times 10^5 \, M^{-1} \, s^{-1}$; $k_{off} = 5.6 \times 10^{-3} \, s^{-1}$; $K_D = 2.2 \times 10^{-8} \, M$) than parent hTAAB ($k_{on} = 1.4 \times 10^5 \, M^{-1} \, s^{-1}$; $k_{off} = 6.0 \times 10^{-4} \, s^{-1}$; $K_D = 4.2 \times 10^{-9} \, M$) (Fig. 6i), indicating that conventional CDR grafting is insufficient to maintain binding affinity of hTAAB to hTie2.

**Fig. 5 Ligand-independent Tie2 dimerization is essential for hTAAB-mediated Tie2 clustering and activation. a** Design of the Tie2 monomeric mutant. Residues V685, V687, and K700 (yellow) at Fn3–Fn3′ dimeric interface were mutated to negatively charged residues (V685D/V687D/K700E) for disrupting the Fn3–Fn3′ dimeric interface through charge repulsions, thus generating a constitutive Tie2 monomer (left). Schematic representation of full-length Tie2-GFP WT, as well as constitutive Tie2 dimeric and monomeric mutants which were used for Live-cell imaging and Tie2-activation assays in HEK293T cells (right). **b** Representative confocal time-lapse images of Tie2 (full-length Tie2-GFP WT, Tie2 dimeric mutant-GFP (D682C/N691C), or monomeric mutant-GFP (V685D/V687D/K700E)) clustering in HEK293T cells after treatment with COMP-Angpt1 (1 μg/ml) or hTAAB IgG1 (10 μg/ml). Similar results were observed in three independent experiments. Images are shown at ×60 magnification. Scale bars, 10 μm. **c** Quantification of the total area of GFP puncta per cell after treatment with COMP-Angpt1 or hTAAB. Each dot indicates a value of the total area of GFP puncta per cell and $n = 10$ cells/group pooled from three independent experiments. Data were analyzed and expressed as mean ± SEM (***$P < 0.001$ vs. control). $P$ values by one-way ANOVA test followed by Tukey's multiple comparisons test. ns, not significant. **d** Fluorescence size-exclusion chromatography (FSEC) analysis of full-length Tie2-GFT WT and dimeric or monomeric Tie2 mutant solubilized from HEK293F membranes by LMNG-CHS. **e, f** Immunoblot analysis for relative phosphorylation ratios of Tie2 and Akt in HEK293T cells transiently expressing WT Tie2, constitutive dimeric Tie2 mutant (D682C/N691C), or constitutive monomeric Tie2 mutant (V685D/V687D/K700E) after treatment with COMP-Angpt1 (1 μg/ml) or hTAAB IgG1 (10 μg/ml) for 1 h (**e**). Densitometric analyses (**f**) of p-Tie2/Tie2 and p-Akt/Akt ratios are shown. Data from three independent experiments ($n = 3$) were analyzed and expressed as mean ± SD (**$P = 0.0035$ (CA1 in dimeric Tie2 group), 0.0019 (hTAAB in dimeric Tie2 group) vs. control in dimeric Tie2 group, ***$P < 0.001$ vs. control for each group). $P$ values by one-way ANOVA test followed by Sidak's multiple comparisons test. ns, not significant. **g** Model for hTAAB-mediated Tie2 activation on cell membrane. hTAAB IgG binds to the Fn3 domain of Tie2 homodimer and induces the clustering of ligand-independent Tie2 dimers into higher-order circular assemblies. Tie2 clustering organizes inactive kinase dimers in a manner that is optimal for autophosphorylation between neighboring dimers.

For structure-based humanization of hTAAB, we performed homology modeling of hzTAAB-H1 and hzTAAB-L1 using the Fv (variable fragment) structure of parental hTAAB in our crystal structure as a template. We then superimposed parent hTAAB Fv structure on the resulting hzTAAB-H1L1 Fv model to identify FR residues that are critical for maintaining VH–VL pairing, CDR conformation, and binding affinity to hTie2 (Fig. 6e). A comparative analysis between the VH–VL interface of hzTAAB-H1L1 and parental hTAAB revealed that the hTAAB light-chain residues N34 and L36 (G34 and Y36 in hzTAAB-L1) on LFR2 can stabilize Y101 on HCDR3 and relieve a steric hindrance with W105 on HFR4. Moreover, hTAAB light-chain residue D55 (Q55 in hzTAAB-L1) on LFR3 is critical for maintaining the conformation of the LCDR2 hairpin loop by interacting with R46 on LFR2 (Fig. 6f). hTAAB light-chain residue R66 (G66 in hzTAAB-L1) on LFR3 also is critical to Tie2 binding through charge interactions with E705 of hTie2 (Fig. 6g). Based on these rationales, we generated hzTAAB-L2 by back-mutating selected residues of hzTAAB-L1 to the corresponding mouse hTAAB residues (G34N, Y36L, Q55D, and G66R) (Fig. 6d). For heavy chain (hzTAAB-H2), we mutated R72 and T74 on HFR3 of hzTAAB-H1 to parent hTAAB residues V72 and K74, which potentially stabilize the HCDR2 conformation by interacting with P53 and S54 of HCDR2 (Fig. 6h).

Following hzTAAB-H1L1, we further produced three other hzTAABs—hzTAAB-H1L2, hzTAAB-H2L1, and hzTAAB-H2L2—according to different combinations of heavy and light chains, all as full-length IgG1s in HEK293F cells (Supplementary Fig. 5d, e). After purification, we evaluated the binding kinetics of the resulting antibodies to hTie2 ECD using SPR. The binding affinity of hzTAAB-H2L1 ($k_{on} = 2.8 \times 10^5\,M^{-1}\,s^{-1}$; $k_{off} = 3.3 \times 10^{-3}\,s^{-1}$; $K_D = 1.1 \times 10^{-8}\,M$) was slightly higher than that of hzTAAB-H1L1 ($k_{on} = 2.5 \times 10^5\,M^{-1}\,s^{-1}$; $k_{off} = 5.6 \times 10^{-3}\,s^{-1}$; $K_D = 2.2 \times 10^{-8}\,M$), but its dissociation rate was still much higher than that of parental hTAAB ($k_{on} = 1.4 \times 10^5\,M^{-1}\,s^{-1}$; $k_{off} = 6.0 \times 10^{-4}\,s^{-1}$; $K_D = 4.2 \times 10^{-9}\,M$) (Fig. 6i). However, back mutations in the light chain significantly improved the affinities of hzTAAB-H1L2 ($k_{on} = 1.1 \times 10^5\,M^{-1}\,s^{-1}$; $k_{off} = 9.6 \times 10^{-4}\,s^{-1}$; $K_D = 8.5 \times 10^{-9}\,M$) and hzTAAB-H2L2 ($k_{on} = 1.4 \times 10^5\,M^{-1}\,s^{-1}$; $k_{off} = 7.3 \times 10^{-4}\,s^{-1}$; $K_D = 5.2 \times 10^{-9}\,M$) for hTie2 ECD (Fig. 6i). Affinities of the resulting humanized antibodies (hzTAAB-H1L2 and hzTAAB-H2L2) were comparable to the affinity of parent hTAAB and were achieved mainly by a significant reduction in dissociation rate. Similar to parental hTAAB, none of these humanized antibodies are bound to mTie2 (Supplementary Fig. 5f). Among these hzTAABs,

hzTAAB-H2L2 most potently induced Tie2 and Akt phosphorylation in HUVECs (Fig. 7a, b). Accordingly, hzTAAB-H2L2 potently induced survival, migration, and tube formation of ECs, enlarged sprouting from EC spheroids, and Tie2 translocation to cell–cell contact sites and FOXO1 translocation from the nucleus to cytosol, doing so to an extent similar to that of COMP-Angpt1 and parental hTAAB (Fig. 7c–f). Moreover, hzTAAB-H2L2 can induce Tie2 and Akt phosphorylation regardless of the presence of Angpt2, suggesting that hTAAB and hzTAAB are resistant to inhibition by antagonistic Angpt2 (Fig. 7g), similar to Tie2-synthetic ligand-1 (TSL1)[28]. To overcome antagonism by Angpt2 in a pathological condition, the level of exogenous Angpt1 variants should be high enough, but this raises the possibility of deleterious effects, such as increased vessel remodeling in established vessels. Our results suggest that hTAAB which is not inhibited by Ang2 could potentially be used at lower concentrations, decreasing the likelihood of side effects. Together, our structure-based humanization strategy successfully retained the binding affinity and Tie2-agonistic activity of parental hTAAB.

## Discussion

In this study, we generate a human Tie2-agonistic antibody, hTAAB, using mouse hybridoma technology and perform structure-based humanization of hTAAB for future clinical applications. Our structural and functional analysis with Tie2 and hTAAB suggests a model for Tie2 clustering, in which polygonal assembly of preformed Tie2 homodimers is mediated by direct binding of hTAAB IgG to the Tie2 Fn3 domain. In addition, we show that this polygonal Tie2 clustering is critically required for hTAAB-mediated Tie2 activation.

The results of our previous preclinical studies with Angpt1 variants and ABTAA have demonstrated that enhancing healthy angiogenesis, vascular integrity, and stability through Tie2 activation provides definitive benefits for treating vascular dysfunctional diseases such as ischemic retinopathy, wet-type aged macular degeneration, sepsis, and primary angle glaucoma, and for enhancing drug delivery through tumor vessel normalization[16–19,29]. In this study, we sought to develop a Tie2-agonistic antibody that free from antagonism or modulation by endogenous Angpt. Therefore, we select the membrane-proximal Ig3–Fn3 domain instead of the LBD of Tie2 for mouse immunization. Subsequent activity-based screenings with hybridoma clones successfully identify potent Tie2-agonistic antibodies. Our structural and functional characterizations of hTAAB demonstrate that, indeed, immunization with Ig3–Fn3 is crucial for generating potent Tie2-agonistic antibodies. The crystal structure

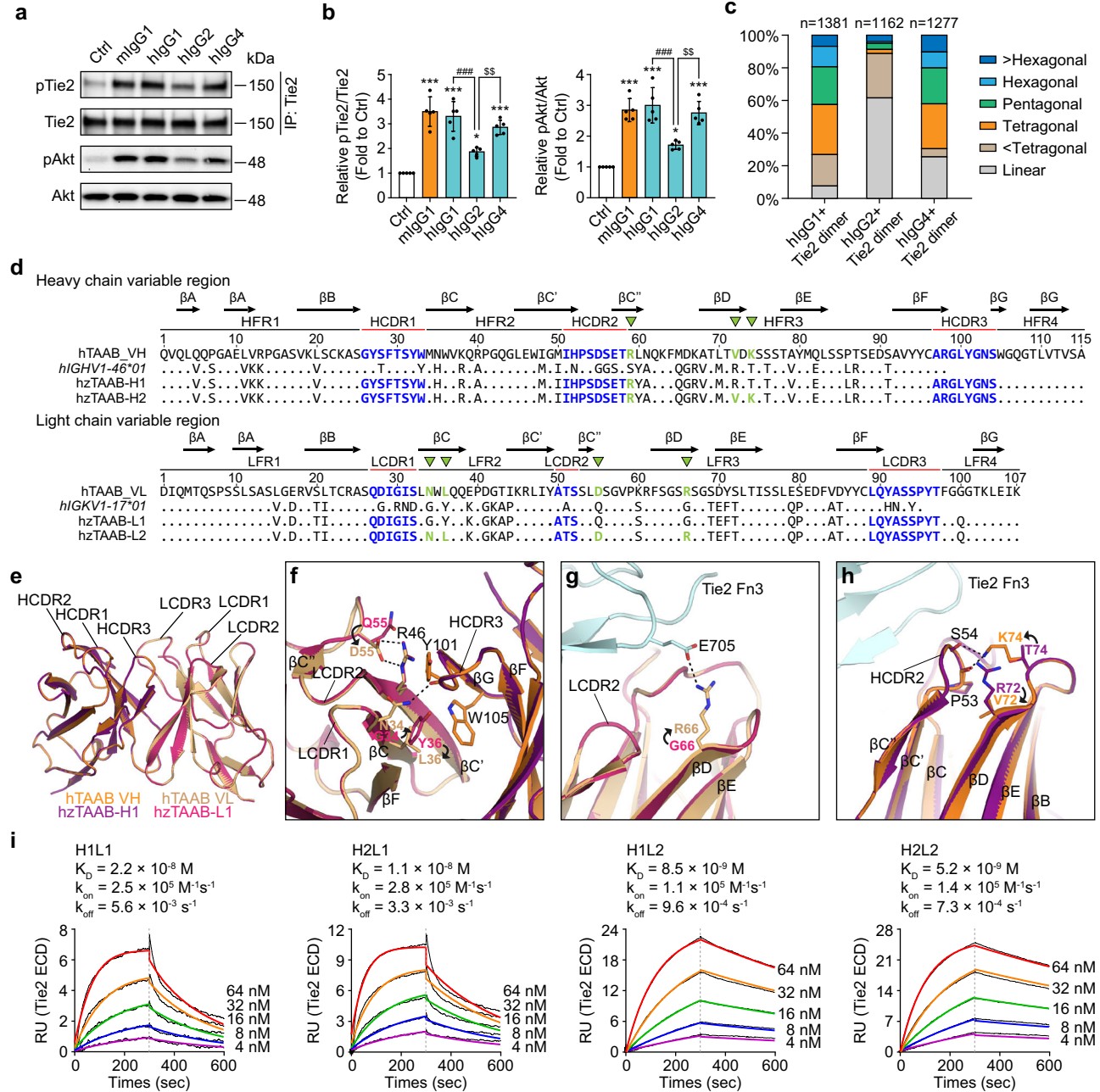

of Tie2 Fn2–3 in complex with hTAAB Fab reveals two Fabs binding to the lateral sides of the Fn3 domains in a V-shaped Tie2 homodimer with 2:2 stoichiometry. Even in the presence of hTAAB Fab, ligand-independent Tie2 homodimerization can be mainly attributed to the intermolecular antiparallel β-sheet between two Fn3 domains of Tie2, which is consistent with previous structures for apo-Tie2 dimers (5MYB and Dimer2 model of 5UTK)[12,13]. Although Moore et al. suggested that the loops in the Fn2/Fn3 boundary region can also mediate Tie2 homodimerization (Dimer1 model of 5UTK), given that epitopes of the Fn3 domain should be exposed to the surface for hTAAB accessibility, Fn3-mediated Tie2 dimerization is a more favorable model.

In general, ligand binding induces dimerization of RTKs, including vascular endothelial growth factor receptor, KIT and Trk, and this ligand-mediated RTK dimerization allows for proper positioning of cytoplasmic kinase domains for autophosphorylation and RTK activation[30]. In contrast, Tie2 in the

cellular membrane exists as a ligand-independent dimer (Tie2 homodimer or Tie2/Tie1 heterodimer), and interaction with high-order oligomeric Angpt1 can trigger activation of these Tie2 dimers[12,13,25,31]. Recent studies have proposed that Fn2 interactions are essential for lateral clustering of preformed Tie2 dimers according to crystal structures and functional analysis with Fn2 mutants[12]. Based on small-angle X-ray scattering and biophysical assays, Leppanen et al. proposed that two of four FLDs in tetrameric Angpt would bind diagonally to neighboring Tie2 monomers within parallel arrays of dimeric Tie2 receptor[32]. Of interest, we can also observe similar interactions between Fn2 domains of neighboring homodimers in the crystal packing (Supplementary Fig. 6a, b), even though our structure contains the Tie2 Fn2–3/hTAAB Fab complex and the space group for the protein crystal differs between the two structures. However, Fab and Fab* bound to Tie2 Fn3 in our crystal-packing lattice are almost parallel to each other (an asterisk is used to denote domains or residues in the symmetric mate of the Tie2 Fn2–3/hTAAB Fab complex),

**Fig. 6 Structure-based humanization of hTAAB. a** Immunoblot analysis of the relative phosphorylation ratios of Tie2 and Akt after treatment of HUVECs with hTAAB in the form of mouse IgG1 or human chimeric IgG1, IgG2, or IgG4 (1 μg/ml). **b** Densitometric analyses of the p-Tie2/Tie2 and p-Akt/Akt ratios. Data from five independent experiments ($n = 5$) were analyzed and expressed as mean ± SD (*$P = 0.0257$ (p-Tie2/Tie2), 0.0397 (p-Akt/Akt) vs. control, ***$P < 0.001$ vs. control; ###$P < 0.001$ hIgG1 vs. hIgG2; $^{$$}P = 0.0078$ (p-Tie2/Tie2), 0.0013 (p-Akt/Akt) hIgG2 vs. hIgG4). $P$ values by one-way ANOVA test followed by Sidak's multiple comparisons test. **c** Quantification of the heterogeneous composition of the Tie2 dimeric mutant/hTAAB IgG subtype complex. The hTAAB IgGs (hIgG1, hIgG2, or hIgG4) were incubated with purified hTie2 Ig3-Fn3 N691C/D682C, followed by size-exclusion chromatography for negative-stain EM analysis. Based on the shape of the complex on the image, we classified the complexes into six categories (linear, <tetragonal, tetragonal, pentagonal, hexagonal, and >hexagonal), and the proportion of each category is indicated (total counted particles $n = 1381$, 1162, and 1277 for IgG1, IgG2, and IgG4, respectively). **d** Alignment of sequences of heavy- and light-chain variable regions of hTAAB and humanized TAAB with the closest human germline gene. Dots in human germline and humanized TAAB sequences indicate residues that are identical to parental hTAAB. CDRs of hTAAB and grafted CDRs in humanized TAAB are colored in blue. Back-mutated residues to parental hTAAB are colored in green and are indicated with green triangles. Secondary structure elements (β-strands) are noted above alignment as arrows. **e** Homology model structure of hzTAAB-H1L1 Fv superimposed on the chimeric hTAAB Fv structure, adapted from the crystal structure of the Tie2 Fn2–Fn3/chimeric hTAAB Fab complex. The heavy and light chains of chimeric hTAAB are colored in dark orange and light orange, respectively, and those of hzTAAB-H1L1 are shown in purple and pink. **f**–**h** Rationales for structure-based humanization of the Tie2-activating antibody. Residues in the light chain critical for maintaining VH–VL pairing and CDR conformation (**f**) as well as affinity for Tie2 Fn3 (**g**) were back-mutated to the mouse hTAAB sequence. Residues in the heavy chain critical for maintaining CDR conformation were back-mutated to the mouse hTAAB sequence (**h**). Mutations are indicated with arrows, and residues are colored the same as in (**e**). **i** Binding kinetics of humanized TAABs (hzTAAB-H1L1, hzTAAB-H2L1, hzTAAB-H1L2, and hzTAAB-H2L2) for hTie2 ECD measured by SPR analysis. The equilibrium dissociation constant ($K_D$, M) was calculated as the ratio of off-rate to on-rate ($k_{off}/k_{on}$). Kinetic parameters were determined with the global fitting function of Biacore Insight Evaluation Software using a 1:1-binding model. Vertical dashed lines indicate the start of the dissociation phase.

thus deviating from the possible angles between two Fab arms of one IgG1 (115–148°)[33]. Accordingly, it is unlikely that hTAAB IgG1 engages the Fn2-mediated lateral clustering of preformed Tie2 dimers, which can be occurred by higher-ordered native Angpt1[32].

Intriguingly, we uncover an unanticipated mode for Tie2 clustering, based on the negative-stain EM analysis of hTAAB IgG1 and constitutive dimeric Tie2 mutant (D682C/N691C) that mimics ligand-independent Tie2 dimers in a physiological cellular membrane. Particularly, the preformed Tie2 dimers are sandwiched between two hTAAB IgG1 molecules, leading to polygonal assemblies (tetragonal, pentagonal, and hexagonal closed-ring shapes). The wide range of angles between the two Fab arms in IgG1 (115–148°)[33], resulting from the flexibility of the IgG1 hinge region, enable hTAAB IgG1 to form multiple polygonal assemblies with Tie2 homodimers. Moreover, the distance between adjacent Fn3 domains cross-linked by hTAAB IgG in 2D class average is ~120 Å, which also closely matches the optimal distance for two paratopes in an antibody (~130 Å)[34]. Notably, we show that hTAAB IgG2 with a rigid hinge mostly induces linear Tie2 clusters rather than polygonal Tie2 clusters, eliciting much less Tie2-agonistic activity compared to the IgG1 and IgG4 subclasses. These results indicate that polygonal Tie2 clustering is required for hTAAB-mediated Tie2 activation. Unlike Tie2 WT and constitutive dimeric mutants, the constitutive monomeric Tie2 mutant (V685D/V687D/K700E) is unable to form a Tie2 cluster on the cell membrane and transduce cellular signaling after COMP-Angpt1 or hTAAB treatment, indicating that ligand-independent Tie2 homodimerization by homotypic Fn3 interactions is crucial for Tie2 clustering and activation.

A previous structural study showed that the Tie2 kinase domain exists in a self-inhibitory conformation in which the nucleotide-binding loop and C-terminal tail containing tyrosine autophosphorylation sites occlude the ATP and substrate binding sites, respectively[35]. In addition, the kinase domains of Tie2 in complex with a Tie2 inhibitor (rebastinib) have been crystallized in a back-to-back dimeric arrangement[36]. These orientations impose a large spatial separation between two kinase active sites of the ligand-independent Tie2 homodimer that is unfavorable for cross-phosphorylation, forcing an inactive status. Therefore, we speculate that hTAAB-mediated Tie2 clustering into polygonal circular assemblies reorganizes the inactive kinase domains

of ligand-independent Tie2 homodimers toward an active position that is optimal for autophosphorylation by neighboring dimers (Fig. 5g and Supplementary Fig. 6c), similar to epidermal growth factor receptor, Eph, and RET[37–40]. Of even greater interest, polygonal Tie2/hTAAB IgG1 assemblies that we observe, as well as the position of LBD in our model structure, raise the possibility that five or six FLDs in COMP-Angpt1 (or highly clustered native Angpt1) might induce Tie2 clusters in a manner similar to hTAAB, despite differences in their Tie2 targeting domain (Supplementary Fig. 6d). According to this model, five hTAABs or one COMP-Angpt1 would tether five Tie2 homodimers, which explains why more hTAAB (10 μg/ml, ~150 kDa) is required to activate Tie2 to an extent similar to COMP-Angpt1 (1 μg/ml, ~170 kDa). Further experiments are needed to ascertain the precise dynamics and geometrical arrangement of Tie2/native Angpt1 complexes and kinase domains on the cell membrane.

With the aim of utility in clinical applications, we perform structure-guided humanization of hTAAB, generating hzTAAB-H2L2 with a target-binding affinity and biological efficacy comparable to those of parental hTAAB. To achieve this outcome, we use conventional CDR grafting together with back mutations of several key FR residues critical for maintaining VH–VL pairing, CDR conformation, and binding affinity for hTie2. However, future work will need to address several issues. Tie2 cellular signaling can be further modulated by Tie1, vascular endothelial protein tyrosine phosphatase, and integrins ($\alpha_5\beta_1$, $\alpha_v\beta_3$, and $\alpha_v\beta_5$)[29,31,41–44]. Furthermore, FLD-based Angpt1 variants and ABTAA can influence integrin- and Tie1/Tie2 heterodimer-mediated signaling. In contrast, hTAAB and hzTAAB-H2L2 are more likely to be independent of integrin and Tie1 because these two antibodies specifically bind to Tie2 homodimers. Given that the epitope for hTAAB and hzTAAB-H2L2 is located at the membrane-proximal Fn3 domain, our Tie2-agonistic antibody would be expected to induce the clustering only of *cis* Tie2 homodimers. Native Angpt1 and COMP-Angpt1, however, engage receptor oligomers on the same cell in *cis* or bridge receptors across cell–cell contacts in *trans*[45,46]. In addition, hTAAB and hzTAAB-H2L2 cannot bind mouse Tie2 and thus would not exhibit cross-reactivity in a mouse model. Therefore, the therapeutic potential of our Tie2-agonistic antibodies— hTAAB and hzTAAB-H2L2—should be further explored in appropriate animal models of disease such as non-human

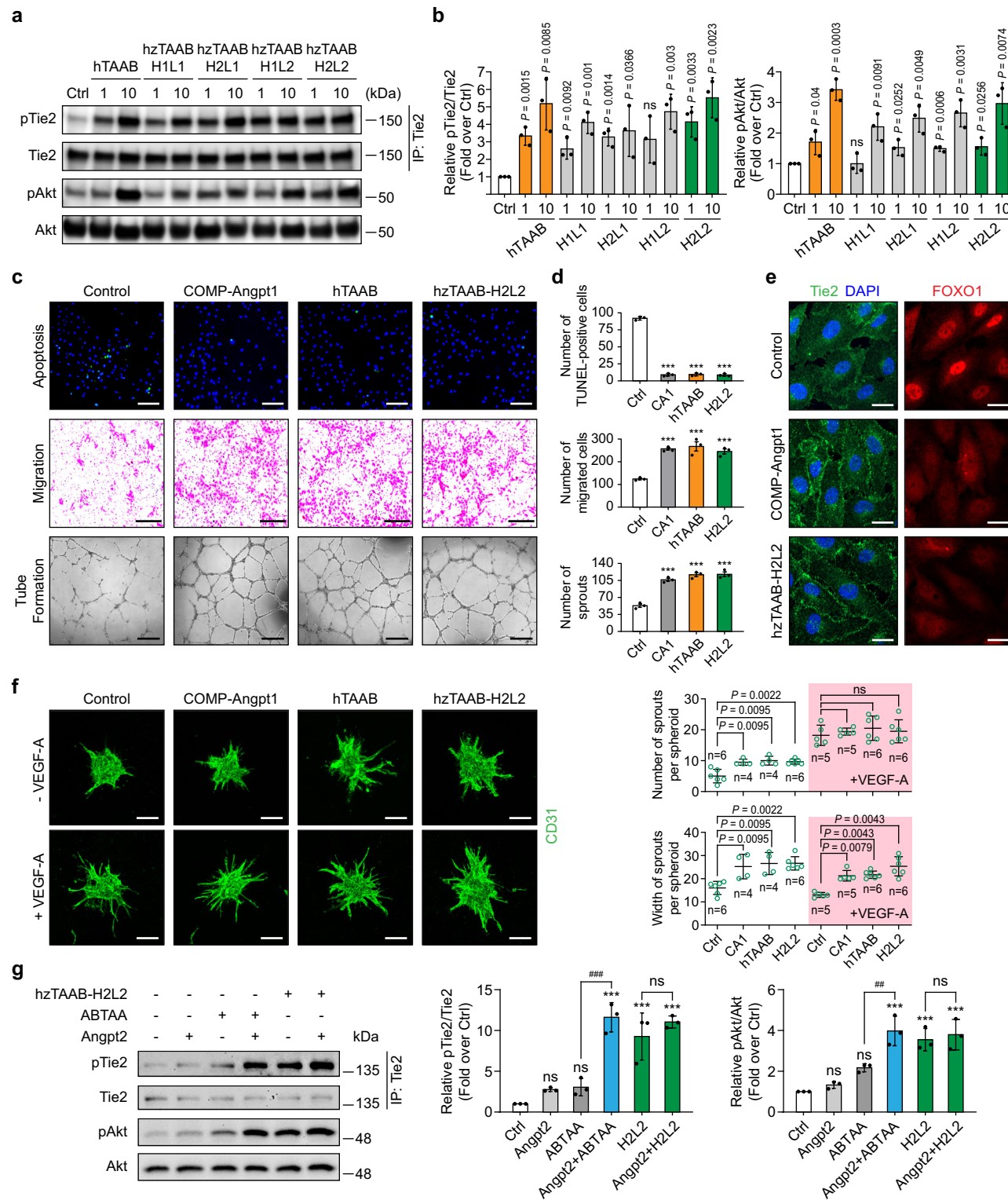

primates and mouse models that express human Tie2. Despite Angpt1 variants, ABTAA and our Tie2-activating antibody (hTAAB) having the same ultimate goal of activating Tie2, they function in slightly different ways, thus likely to lead to somewhat different therapeutic outcome under pathological condition[47]. Therefore, it will be of interest in future studies to examine the therapeutic effects of hzTAAB under pathological conditions in which Angpt2 is elevated and to compare the effect of hzTAAB to that of other Tie2 agonists (COMP-Angpt1, ABTAA, etc.).

Moreover, combined treatment of hzTAAB with Angpt2-blocking antibody can be also investigated.

## Methods

**Cell lines, cell culture, and reagents**. Human umbilical vein endothelial cells (HUVECs; C2519A, Lonza) were maintained in EGM-2 endothelial growth medium (CC-3162; Lonza) and cultured at 37 °C in a humidified 5% $CO_2$ incubator. Expi293F cells (A14527, ThermoFisher) were maintained in Expi293 Expression Medium (A1435102, ThermoFisher) and cultured in a humidified shaking

**Fig. 7 Humanized TAAB activates Tie2 and its downstream signaling in HUVECs. a, b** Immunoblot analysis for relative phosphorylation ratios of Tie2 and Akt after treatment of HUVECs with different concentrations (1 and 10 μg/ml) of humanized TAABs (hzTAAB-H1L1, H1L2, H2L1, H2L2) (**a**). Densitometric analyses (**b**) of p-Tie2/Tie2 and p-Akt/Akt ratios are shown. Data from three independent experiments ($n = 3$) were analyzed and expressed as mean ± SD. $P$ values by two-tailed unpaired $t$ test. ns, not significant. **c, d** Biological effects of hzTAAB-H2L2 in HUVECs. Effect of hzTAAB-H2L2 on serum deprivation-induced apoptosis in HUVECs (Top). HUVECs were incubated with serum-free medium containing 10 μg/ml of hzTAAB-H2L2, hTAAB, or 1 μg/ml of COMP-Angpt1. Migration activity was measured using a modified Boyden chamber assay (Middle). HUVECs were seeded in the upper layer of 8-μm pore membranes. Serum-free medium containing 10 μg/ml of hzTAAB-H2L2, hTAAB, or 1 μg/ml of COMP-Angpt1, was added to the bottom chamber. Migrated cells were fixed and stained after 9-h incubation. HUVECs were plated on Matrigel-coated wells and incubated with serum-free medium containing 10 μg/ml of hzTAAB-H2L2, hTAAB, or 1 μg/ml of COMP-Angpt1. Phase-contrast microscopy images were acquired 7 h after treatment. Tube-formation activity was measured as total tube length (Bottom). Scale bars, 100 μm. Quantification (**d**) of TUNEL-positive cell, migrated HUVEC, and sprouts for (**c**). Data from three independent experiments ($n = 3$) were analyzed and expressed as mean ± SEM (***$P < 0.001$ vs. control). $P$ values by one-way ANOVA test followed by Sidak's multiple comparisons test. **e** Representative images of HUVECs showing hzTAAB-H2L2-induced Tie2 translocation to cell–cell contacts and hzTAAB-H2L2-induced nuclear clearance of FOXO1. Serum-starved HUVECs were treated with hTAAB-H2L2 (10 μg/ml) or COMP-Angpt1 (1 μg/ml) for 30 min. No treatment was used as a negative control. Goat anti-human Tie2 and Alexa Fluor 488-conjugated donkey anti-goat antibodies were used for Tie2 visualization, and rabbit anti-FOXO1 and Alexa Fluor 594-conjugated donkey anti-rabbit antibodies were used for FOXO1 visualization. Scale bars, 20 μm. DAPI, 4′, 6-diamidino-2-phenylindole. Similar results were observed in three independent experiments. **f** Representative images of HUVEC spheroids 3D culture after stimulation with COMP-Angpt1 (1 μg/ml), hTAAB (10 μg/ml), or hzTAAB (10 μg/ml) with or without VEGF-A for 48 h (left). Scale bars, 100 μm. Comparison of the number and width of sprouts is shown in the right panel and each dot indicates a value from $n = 4$–6 spheroids/group (mean ± SD). $P$ values by two-tailed Mann–Whitney $U$ test. **g** Immunoblot analysis of the relative phosphorylation ratios of Tie2 and Akt after treatment of HUVECs with ABTAA or hzTAAB-H2L2 (10 μg/ml) with or without Angpt2 (1 μg/ml). Densitometric analyses of p-Tie2/Tie2 and p-Akt/Akt ratios are shown in the right panel. Data from three independent experiments ($n = 3$) were analyzed and expressed as mean ± SD (***$P < 0.001$ vs. control; ##$P = 0.0051$ ABTAA vs. Angpt2+ABTAA; ###$P < 0.001$ ABTAA vs. Angpt2+ABTAA). $P$ values by one-way ANOVA test followed by Sidak's multiple comparisons test. ns, not significant.

incubator at 37 °C and 8% $CO_2$. B-cell hybridomas were maintained in Dulbecco's Modified Eagle Medium (DMEM) supplemented with 10% fetal bovine serum (FBS), penicillin (100 U/ml), and streptomycin (100 mg/ml). To produce anti-Tie2 antibody from B-cell hybridoma, the cells were washed with PBS and then cultured in serum-free medium (SFM, 12045-076, Gibco) for 3 days.

**Expression and purification of recombinant proteins.** Recombinant Tie2 protein for mouse immunization was generated by subcloning the gene encoding hTie2 Ig3–Fn3 (residues 349–738 of human Tie2, GenBank accession number: AAH35514.2) into the pFuse-hIgG1-Fc vector (pFuse-hg1fc1, InvivoGen) and transiently expressing it in Expi293F cells. Specifically, Expi293F cells ($2 \times 10^6$ cells/ml) were cultured in Expi293 Expression Medium (A1435103, ThermoFisher), and the plasmid encoding hTie2 Ig3–Fn3 was transfected into Expi293F cells using an ExpiFectamine 293 Transfection kit (A14524, ThermoFisher). Cells were cultured at 37 °C under 8% $CO_2$ for 5 days in a shaking incubator (orbital shaker, 125 rpm). After centrifugation to remove the cells, the culture supernatant containing secreted hTie2 Ig3–Fn3–Fc fusion protein was purified with an AKTA purification system (GE Healthcare) equipped with HiTrap MabSelect SuRe affinity column (11003494, GE Healthcare). The purified hTie2 Ig3–Fn3–Fc fusion protein was concentrated using Amicon Ultra centrifugal filter (UFC8030, Millipore) and the buffer was replaced with PBS. The Fc-tag of the fusion protein was removed by thrombin digestion (27-0846-01, GE Healthcare) at 22 °C for 18 h. The hTie2 Ig3–Fn3 protein was further purified by removing the cleaved Fc-tag with HiTrap MabSelect SuRe affinity column.

To prepare human Tie2 for crystallization, the gene encoding the hTie2 Fn2–3 (residues 541–735) was amplified by PCR using primers (Supplementary Table 1) containing *Nde*I and *Xho*I sites for cloning. The PCR product was then subcloned into pET-28a (69864, Novagen) and expressed in *E.coli* BL21 (DE3) RIL (230240, Agilent Technologies). Cells were grown at 37 °C in LB medium supplemented with 50 μg/ml kanamycin to an $OD_{600}$ of 0.4. Protein expression was induced with 0.05 mM IPTG (isopropyl β-d-1-thiogalactopyranoside) and incubated for 15 h at 18 °C. The cells were harvested by centrifugation, re-suspended in lysis buffer (20 mM HEPES pH 7.5, 200 mM NaCl), and lysed by sonication on ice. After the removal of cell debris by centrifugation ($13,000 \times g$ for 0.5 h at 4 °C), the supernatants were applied to an Ni-NTA agarose affinity column (30210, QIAGEN). After washing with five column volumes of wash buffer (20 mM HEPES pH 7.5, 200 mM NaCl, 50 mM imidazole), protein was eluted with elution buffer (20 mM HEPES, pH 7.5, 200 mM NaCl, 400 mM imidazole) and further purified by size-exclusion chromatography using a HiLoad 16/600 Superdex 200 pg column (28-9893-35, GE Healthcare) equilibrated with 20 mM HEPES pH 7.5 and 200 mM NaCl.

To prepare chimeric hTAAB Fab for crystallization, the synthesized heavy and light chains for chimeric hTAAB Fab were subcloned into a modified pBAD expression vector using primers (Supplementary Table 1) for periplasmic secretion[48]. *E.coli* Top10F (Invitrogen) cells were transformed with the plasmid pBAD-chimeric hTAAB Fab, and then grown at 37 °C in LB medium supplemented with 100 μg/ml ampicillin. At an $OD_{600}$ of 1.0, protein expression was induced with 0.2% arabinose, and cells were grown at 30 °C for 15 h. The cells were harvested by centrifugation, re-suspended in lysis buffer (20 mM HEPES pH 7.5, 200 mM NaCl), and lysed by sonication on ice. After removal of cell debris by

centrifugation ($13,000 \times g$ for 30 min at 4 °C), supernatants containing soluble chimeric hTAAB Fab were applied to Ni-NTA agarose affinity chromatography columns (QIAGEN) and washed with five column volumes of wash buffer (20 mM HEPES pH 7.5, 200 mM NaCl, 50 mM imidazole). The protein was then eluted with elution buffer (20 mM HEPES pH 7.5, 200 mM NaCl, 400 mM imidazole) and further purified by size-exclusion chromatography using a HiLoad 16/600 Superdex 200 pg column equilibrated with 20 mM HEPES pH 7.5 and 200 mM NaCl. Purified proteins and antibodies were aliquoted and stored at −80 °C.

**Immunization and generation of B-cell hybridomas.** All animal studies involving mouse immunization, euthanasia, and organ harvesting were conducted by AbClon Inc. (Seoul, Korea) in accordance with the guidelines of the National Institute of Health (NIH) "Guide for Care and Use of Animals" and an approved protocol received by the company's Institution Animal Care and Use Committee. Five-week-old BALB/c mice were immunized twice weekly for 6 weeks with purified hTie2 Ig3–Fn3 (100 μg/injection), mixed with an adjuvant. Anti-Tie2 antibody titers in sera of immunized mice were assessed by hTie2 enzyme-linked immunosorbent assay (ELISA). When antibody titers (at a 1:5000 dilution) suitably increased (OD > 1.0), B lymphocytes were isolated from spleens of immunized mice and fused with cultured myeloma cells (SP2/0). The fused cells were cultured in HAT (hypoxanthine, aminopterin, and thymidine) medium, and hybridoma cells comprising only fused myeloma cells and B lymphocytes were selected and cultured[49]. Surviving hybridoma cells were seeded in 96-well plates, and culture supernatants were tested by hTie2 ELISA. Hybridoma pools exhibiting positive signals were chosen for clonal selection through limiting dilution.

**ELISA.** For hTie2 ELISA, 96-well plates (Corning) were coated with hTie2 Ig3–Fn3 (100 ng/well) by incubating overnight at 4 °C in 50 mM carbonate buffer (pH 9.6), followed by blocking with PBS containing 3% (w/v) skim milk for 1 h at 25 °C. After washing with PBS, 100 μl of hybridoma cell culture supernatant was added to each well and the plates were incubated for 2 h at 25 °C. After washing with PBS, bound antibodies were detected by labeling with horseradish peroxidase (HRP)-conjugated anti-mouse IgG antibody (1:10,000 dilution, HAF007, R&D systems) and subsequently incubating with TMB (3,3′,5,5′-tetramethylbenzidine) solution (34028, ThermoFisher). Absorbance was read at 450 nm on a Victor X3 microplate reader (PerkinElmer). The BSA-coated background control value was subtracted in the presented results.

**Screening of positive hybridoma clones by analyzing Akt phosphorylation.** HUVECs ($1 \times 10^5$ cells/ml) were cultured in EGM-2 medium (Lonza) at 37 °C in 60-mm culture dishes. Cells at ~90% confluency were serum-starved by incubating with serum-free EBM-2 medium for 6 h, then treated with anti-Tie2 antibody candidates and further incubated for 30 min. The cells were washed with cold PBS and lysed with lysis buffer (10 mM Tris-Cl pH 7.4, 150 mM NaCl, 5 mM EDTA, 10% glycerol, 1% Triton X-100, protease inhibitor, phosphatase inhibitor) at 4 °C for 20 min. Then, the cell lysates were prepared by centrifugation at $18,000 \times g$ for 15 min, treated with sodium dodecyl sulfate (SDS) sample buffer, and subjected to denaturing polyacrylamide gel electrophoresis (SDS-PAGE). Proteins were

transferred to a nitrocellulose membrane (GE Healthcare). Akt phosphorylation was assessed by first blocking the blot by incubating with 5% (w/v) skim milk in Tris-buffered saline containing 0.1% Tween-20 (TBS-T) at room temperature (RT) for 1 h, and then incubating with an anti-phospho-Akt antibody (S473) (1:1000 dilution, 4058, Cell Signaling) at 4 °C for 8 h, followed by incubation with HRP-conjugated goat anti-rabbit IgG antibody (1:5000 dilution, 12-348, Sigma-Aldrich). Immunoreactive phospho-Akt was visualized using enhanced chemiluminescence (ECL). After staining, the membrane was incubated in stripping buffer (Thermo) for 15 min and re-probed with a rabbit anti-Akt antibody (1:1000 dilution, 9272, Cell Signaling), followed by incubation with HRP-conjugated goat anti-rabbit IgG antibody (1:5000 dilution, 12-348, Sigma-Aldrich) to determine the amount of total Akt.

**Immunoprecipitation of Tie2 and analysis of Tie2 phosphorylation.** HUVECs (Lonza) were cultured in EGM-2 (Lonza) at 37 °C and 5% $CO_2$ in 100-mm culture dishes. At ~80–90% confluence, cells were starved by incubating in EBM-2 (Lonza) medium for 6 h, after which they were treated with different concentrations of anti-Tie2 antibodies (hTAAB IgG1, chimeric hTAAB Fab, different isotypes of hTAAB, or humanized versions) for 30 min. Thereafter, cells were washed twice with cold PBS, lysed in 1 ml of lysis buffer (10 mM Tris-Cl pH 7.4, 150 mM NaCl, 5 mM EDTA, 10% glycerol, 1% Triton X-100, protease inhibitor, phosphatase inhibitor), and then incubated at 4 °C for 60 min. Cell lysates were cleared by centrifugation at 15,000 × g for 10 min, after which the protein concentration in the supernatant was quantified by BCA assay. For Tie2 immunoprecipitation, 1 μg of anti-Tie2 antibody (AF313, R&D Systems) was added to 0.5 mg of lysates and incubated overnight at 4 °C with shaking. Dynabeads Protein G (Life Technologies) was then added and allowed to react for 2 h. The beads were immobilized on one side of the tube using a magnet, washed three times with lysis buffer, and then incubated at 70 °C for 10 min with 2× SDS sample buffer containing a reducing agent (2-mercaptoethanol). The sample was separated from the beads, electrophoresed on a 4–15% SDS protein gel (Bio-Rad), and transferred to a 0.45-μm PVDF membrane. The membrane was blocked by incubating with 5% BSA in TBS-T at RT for 1 h and then incubated with mouse anti-phospho-tyrosine antibody (1:1000 dilution, 4G10, Millipore) at 4 °C for 8 h, followed by incubation with HRP-conjugated goat anti-mouse IgG antibody (1:5000 dilution, HAF007, R&D Systems) and subsequent western blot analysis. The amount of immunoprecipitated Tie2 was measured by incubating the membrane in stripping buffer (Thermo) for 15 min, then blocking again and re-probing with an anti-Tie2 antibody (1:1000 dilution, AF313, R&D Systems), followed by incubation with HRP-conjugated donkey anti-goat IgG antibody (1:5000 dilution, HAF109, R&D Systems).

**Immunofluorescence assays.** Effects of anti-Tie2 antibody on Tie2 localization and FOXO1 translocation from the nucleus to cytosol were examined by immunofluorescence staining in HUVECs. Specifically, HUVECs were seeded on eight-well slide chambers (Lab-TekII) and maintained in EGM-2 medium for 2–3 days. When cells reached 100% confluence, they were serum-starved by incubating in EBM-2 medium for 4 h; then they were treated with 10 μg/ml of hTAAB or 1 μg/ml of COMP-Angpt1 for 30 min. Thereafter, cells were fixed by incubating with 4% formaldehyde in PBS at RT for 10 min, permeabilized with 0.1% Triton X-100 in PBS, blocked with 1% BSA in PBS at RT for 60 min, and incubated at RT for 1 h with primary antibodies [goat anti-human Tie2 (1:200 dilution, AF313, R&D Systems) or rabbit anti-FOXO1 (C29H4) (1:400 dilution, 2880, Cell Signaling)]. The cells were then incubated with Alexa Fluor 488-conjugated donkey anti-goat (1:250 dilution, 705-546-147, Jackson ImmunoResearch) or Alexa Fluor 594-conjugated donkey anti-rabbit (1:250 dilution, 711-585-152, Jackson ImmunoResearch) secondary antibodies in the dark at RT for 1 h and mounted with Vectashield mounting medium containing 4′,6-diamidino-2-phenylindole (DAPI; Vector Labs). Images were acquired with a laser-scanning confocal microscope (LSM880, Carl Zeiss).

**Expansion of hybridoma clones and production of monoclonal antibodies.** In order to purify the anti-Tie2 antibody candidates from ELISA-positive clones, hybridoma cells were cultured in 10% FBS-containing DMEM in a T75 flask. When cells reached ~90% confluence, they were washed with PBS, after which the medium was replaced with 50 ml of serum-free medium (SFM; Gibco) and cells were cultured at 37 °C for 3 days. Culture supernatants containing secreted antibodies from each monoclonal hybridoma were then centrifuged to remove the cells and filtered. The antibodies were purified using an AKTA purification system (GE Healthcare) equipped with a Protein G affinity column (GE Healthcare). Bound antibodies were eluted with IgG Elution Buffer (21009, ThermoFisher) and neutralized with 1 M Tris-Cl (pH 9.0). Purified antibodies were concentrated using an Amicon Ultra centrifugal filter (UFC8030, Millipore), with concurrent replacement of buffer with PBS.

**Sequencing of DNA encoding monoclonal antibodies.** After culturing hybridoma cells (2 × 10$^6$ cells/ml) in 10% FBS-containing DMEM, total RNA was obtained using an RNeasy mini kit (Qiagen). The RNA concentration was measured, and cDNA was synthesized by reverse transcription (RT). Heavy- and light-chain variable region gene sequences were amplified by PCR using a Mouse Ig-

Primer set (69831-3, Novagen) and the synthesized cDNA as a template under the following conditions: 94 °C for 5 min, followed by 35 cycles of 94 °C for 1 min, 50 °C for 1 min, and 72 °C for 2 min, and then 72 °C for 6 min with gradual cooling to 4 °C. The PCR product obtained from each reaction was cloned into a TA vector and subjected to DNA sequencing, yielding nucleotide sequences encoding variable heavy- and light-chain regions of each antibody.

**Crystallization and structural determination of Tie2 Fn2–3/chimeric hTAAB Fab complex.** Purified chimeric hTAAB Fab and hTie2 Fn2–3 were mixed in a 1.2:1 molar ratio and incubated for 1 h at 4 °C. To obtain chimeric hTAAB Fab in complex with hTie2 Fn2–3 and remove unbound Fab, we applied the mixture to a size-exclusion column (HiLoad 16/600 Superdex 200 pg) equilibrated with 20 mM HEPES, pH 7.5, and 200 mM NaCl. Fractions containing the hTie2 Fn2–3/chimeric hTAAB Fab complex were concentrated to 10 mg/ml for crystallization. Crystals were grown using the hanging-drop vapor diffusion method with a reservoir solution containing 100 mM Bis-Tris propane pH 8.5, 110 mM potassium dihydrogen phosphate, and 15% (w/v) polyethylene glycol 3350 at 19 °C. Crystals were obtained within 1 week. For data collection at 100 K, crystals were transferred to a crystallization solution supplemented with 30% ethylene glycol and flash-frozen in liquid nitrogen.

X-ray diffraction data were collected at the beamline 11C of the Pohang Accelerator Laboratory, Republic of Korea. Diffraction data were indexed, integrated, scaled, and merged using HKL2000 (HKL Research)[50], and initial phases were calculated with the molecular replacement technique using Phaser (v2.8.3) with the durvalumab Fab structure (PDB: 5X8M) and Tie2 Fn2–3 structure (PDB: 5MYA) as search probes[12,24]. Iterative rounds of model-building and refinement were done using COOT (v0.8.9.2) and PHENIX (v1.19.2-4158), respectively[51,52]. All structure figures were generated using PyMol (v2.3.2)[53] and ChimeraX (v0.91)[54].

**Size-exclusion chromatography-multiangle light scattering (SEC-MALS).** Protein samples were concentrated to 3 mg/ml for hTie2 Fn2–3 or 1 mg/ml for hTAAB IgG1, hTie2 ECD, and hTAAB IgG1/hTie2 Fn2–3 and hTAAB IgG1/hTie2 ECD complexes. SEC-MALS was carried out at RT using a chromatographic system consisting of a Shimadzu HPLC system and DAWN HELEOS-II (Wyatt Technology). For chromatographic separations, a Superdex 200 Increase 10/300 GL column (28-9909-44, GE Healthcare) was used at an eluent flow rate of 0.5 ml/min with 20 mM HEPES, pH 7.5, and 200 mM NaCl buffer. The weight-average molar mass was calculated using ASTRA v6 software (Wyatt Technology).

**Formation of Tie2 dimer/hTAAB IgG1 complexes.** The hTie2 Ig3–Fn3 D682C/N691C construct was generated by overlap extension PCR using the pFuse-hTie2 Ig3–Fn3–hIgG1–Fc construct. Recombinant hTie2 Ig3–Fn3 D682C/N691C protein was expressed and purified as described above for recombinant hTie2 Ig3–Fn3. Purity and dimeric state were evaluated by SDS-PAGE and Coomassie blue staining under reducing and non-reducing conditions. Purified hTie2 Ig3–Fn3 D682C/N691C was incubated with hTAAB IgG1 for 2 h on ice at a molar ratio of 2:1 (Tie2 monomer:hTAAB IgG1) and isolated by size-exclusion chromatography using a Superose 6 Increase 10/300 GL column (29-0915-96, GE Healthcare) equilibrated with 20 mM HEPES, pH 7.5 and 200 mM NaCl. Elution fractions were analyzed by SDS-PAGE and Coomassie blue staining under reducing and non-reducing conditions.

**Negative-stain electron microscopy and image processing.** SEC fractions containing hTie2 Ig3–Fn3 D682C/N691C in complex with hTAAB IgG1 were diluted to 0.01 mg/ml in 20 mM HEPES, pH 7.5, 200 mM NaCl, and 2.5 μl of the protein was placed onto a glow-discharged CF300-CU grid (Electron Microscopy Sciences). After a 30-s incubation, the sample was blotted with Whatman paper, dipped two times into 20 μl of distilled water, and stained twice with 20 μl of a 0.76% uranyl formate solution for 1 min. Electron micrographs were collected at ×52,000 magnification on a Tecnai G2 spirit microscope (FEI) at 120 kV with a 4 K × 4 K Eagle camera (FEI), using a −1.8 μm defocus value and a dose of 30 e−/Å$^2$. The resulting pixel size was 2.11 Å per pixel. Micrographs were imported into cryoSPARC[55], and contrast transfer function parameters were calculated using patch CTF estimation. A total of 2579 particles were manually selected from 38 micrographs, and 2D class averages were generated after extracting the particles with a 320-pixel box and binning to 160 pixels. 2D classification was run in cryoSPARC with default settings, except the number of 2D classes was set to 8. Objectively good (showing clear Fc and Tie2 density) class averages were selected for further analysis.

**Live-cell imaging and Tie2 activation assays in HEK293T cells.** For expression of full-length Tie2 in HEK293T cells, which lack endogenous Tie receptors[31], the gene encoding full-length human Tie2 (residues 1–1124) and a C-terminal eGFP connected by the nonadhesion linker RSAIT (Arg–Ser–Ala–Ile–Thr) and a Myc epitope tag were subcloned into HindIII and XhoI restriction sites of the pcDNA3.1+ vector. Constitutive dimeric Tie2 (D682C/N691C) and constitutive monomeric Tie2 (V685D/V687D/K700E) mutants were constructed from WT full-length Tie2-linker-Myc-eGFP using QuikChange PCR methods.

HEK293T cells were transiently transfected with WT or mutant Tie2 using Lipo-fectamine 2000 (Invitrogen) and then were incubated for 24 h. For live-cell imaging, cells were treated with 1 µg/ml of COMP-Angpt1 or 10 µg/ml of hTAAB. Cells were imaged every 30 s for 7.5 min with a Nikon A1R confocal microscope (Nikon Instruments) and 488 nm laser at ×60 magnification. The environmental conditions were maintained at 37 °C and 10 or 5% $CO_2$ (Live Cell Instruments) using a Chamlide TC system. The images were analyzed with Nikon imaging software (NIS-element AR 64-bit version 3.10; Laboratory Imaging) and ImageJ (v1.52) software. For quantification of Tie2 clustering, we defined a cluster as discrete GFP puncta with fluorescence intensity of 50–255 a.u. and size >0.2 µm EqDiameter and measured the total area for GFP puncta per cell in ten different cells for each condition using the "Objective Count" tool in the Nikon imaging software. For Tie2-activation assays, confluent cells were serum-starved for 12 h and incubated with 1 µg/ml of COMP-Angpt1 or 10 µg/ml of hTAAB for 1 h. Tie2 and Akt phosphorylation were analyzed using cell lysates, as indicated above.

**Generation of chimeric antibody.** Human–mouse chimeric IgG1, IgG2, and IgG4 antibodies of Tie2-activating antibody hTAAB were generated by cloning the heavy or light-chain variable regions (VH or VL) of mouse hTAAB into backbone vectors containing the human heavy or light-chain constant regions (CH or CL), respectively. In detail, a DNA fragment encoding the heavy-chain variable region of mouse hTAAB was synthesized (Bioneer, Inc.) as the sequence, "EcoRV-signal sequence-VH-NheI". The synthesized DNA fragment was digested with EcoRV and NheI and then sub-cloned into pFUSE-CHIg-hG1 (IgG1 isotype) and pFUSE-CHIg-hG2 (IgG2 isotype) vectors, respectively (InvivoGen). For the production of the human IgG4 chimeric antibody, DNA encoding the VH of mouse hTAAB was amplified by PCR using primers (Supplementary Table 1). The PCR product was then subcloned between EcoRI and NheI sites of a modified pOptiVEC-TOPO vector containing the CH of human IgG4 antibody. The chimeric light-chain expression vector was constructed by PCR-amplifying the DNA fragment encoding the light-chain variable region of mouse hTAAB using primers containing EcoRI and BsiWI restriction sites (Supplementary Table 1). The PCR product was then subcloned into a modified pcDNA3.3-TOPO vector containing the constant region of the human kappa light chain. Chimeric antibodies were prepared using the Expi293 expression system (ThermoFisher). Briefly, Expi293F cells were co-transfected with heavy-chain and light-chain expression vectors using the ExpiFectamine 293 Transfection kit (A14524, ThermoFisher), after which transfected cells were cultured for 5 days in Expi293 expression medium. The culture supernatant was filtered with a 0.45-µm filter and antibodies were purified using an AKTA purification device (GE Healthcare) equipped with a HiTrap Mab-Select SuRe column (11003494, GE Healthcare).

**Humanized antibody preparation.** The Fv region of mouse hTAAB was humanized based on the Tie2 Fn2–3/chimeric hTAAB Fab complex structure and the model structure of IGHV1-46*01 and IGKV1-17*01 complexes. Briefly, the mouse hTAAB Fv sequence was compared with that of human germline genes using the IMGT/DomainGapAlign online tool[27] from the International ImmunoGeneTics information system (IMGT) (http://www.imgt.org). IGHV1-46*01 and IGKV1-17*01, which exhibited the highest sequence identity to heavy- and light-chain variable regions of mouse hTAAB, respectively, were selected as human framework (FR) donors. The CDRs (defined by IMGT numbering) of IGHV1-46*01 and IGKV1-17*01 were replaced with their counterparts in mouse hTAAB, yielding hzTAAB-H1 and hzTAAB-L1. Selected residues critical for maintaining VH–VL pairing, CDR conformation, and affinity for Tie2 Fn3 were further substituted onto hzTAAB-H1 and hzTAAB-L1, generating hzTAAB-H2 and hzTAAB-L2. The resulting humanized Fv genes—hzTAAB-H1, hzTAAB-H2, hzTAAB-L1, and hzTAAB-L2—were synthesized (Bioneer) and optimized for expression in mammalian cells. Constant light-chain regions of human immunoglobulin kappa (GenBank accession number: AAA58989.1) and heavy-chain regions of human immunoglobulin gamma 1 (GenBank accession number: AWK57454.1) were first cloned into pOptiVEC-TOPO and pcDNA3.3-TOPO vectors, respectively, into which the synthesized variable regions of the humanized light chain and heavy chain (hzTAAB-H1, hzTAAB-H2, hzTAAB-L1 or hzTAAB-L2) were subsequently cloned. Plasmids containing humanized light chains and heavy chains were transfected into Expi293F cells using an ExpiFectamine 293 Transfection kit (A14524, ThermoFisher). Cells were cultured at 37 °C under 8% $CO_2$ for 5 days in a shaking incubator (orbital shaker, 125 rpm). The harvested culture supernatants were centrifuged to remove cells, and antibodies were isolated by affinity chromatography on a ProA column (Amicogen) equilibrated in PBS. Bound antibodies were eluted with 0.1 M glycine-HCl, pH 2.7, and neutralized with 1 M Tris-Cl, pH 9.0. Eluted antibodies were further purified by size-exclusion chromatography using a Superdex 200 Increase 10/300 GL column (28-9909-44, GE Healthcare) equilibrated in PBS. Antibody purity was evaluated using reducing and non-reducing SDS-PAGE, and purified antibodies were aliquoted and stored at −80 °C.

**Homology modeling of humanized Tie2-activating antibody.** SWISS-MODEL homology modeling was performed with sequences of human germline IGHV1-46*01 and IGKV1-17*01 grafted with hTAAB CDRs using the structure of hTAAB Fv in the Tie2 Fn2–3/chimeric hTAAB Fab complex structure as a template[56]. The resulting model with the highest QMEAN-Z (Qualitative Model Energy ANalysis-Z) score (−0.37) was aligned to the hTAAB Fv structure for further structure-based humanization.

**Determination of binding kinetics between Tie2 antibody and recombinant Tie2 protein.** The binding kinetics of mouse hTAAB and humanized IgG1s for human Tie2 were measured by surface plasmon resonance (SPR) using a Biacore T200 system equipped with certified-grade CM5 series S sensor chips (BR100399, GE Healthcare). HEPES-buffered saline (0.01 M HEPES, 0.15 M NaCl) containing 3 mM ethylenediaminetetraacetic acid (EDTA) and 0.05% (v/v) P20 detergent (HBS-EP + ) was used as reaction and running buffer (BR100669, GE Healthcare). Human Tie2 ECD (produced in house; residues 23–735 of human Tie2, GenBank accession number: AAH35514.2) was immobilized on the surface of a CM5 sensor chip by amine coupling using 10 mM acetate, pH 5.5. Thereafter, mouse hTAAB and humanized IgG1s, diluted in HBS-EP + buffer, were applied over antigen-immobilized sensor chips at 7 different concentrations (0, 2, 4, 8, 16, 32, and 64 nM) for 300 s at a flow rate of 30 µl/min. Analytes bound to sensor chips were dissociated by washing with HBS-EP + running buffer for 300 s. Association ($k_{on}$, $M^{-1} s^{-1}$) and dissociation ($k_{off}$, $s^{-1}$) constants were both measured over 300-s intervals. The equilibrium dissociation constant ($K_D$, M) was calculated as the ratio of off-rate to on-rate ($k_{off}/k_{on}$). Kinetic parameters were determined with the global fitting function of Biacore Insight Evaluation Software using a 1:1-binding model.

The binding kinetics of mouse hTAAB and humanized IgG1s for mouse Tie2 were measured by Biolayer Interferometry (BLI) using the Octet system (ForteBio). Assays were performed at 30 °C. After hydrating in kinetics buffer (0.01% endotoxin-free BSA, 0.002% Tween-20, 0.005% $NaN_3$ in PBS) for 10 min, mouse hTAAB (10 µg/ml each) were loaded onto anti-mouse IgG Fc capture (AMC) biosensors (ForteBio). Association curves were recorded for 600 s by incubating Ab-coated sensors with a 1600 nM solution of mouse Tie2 Ig3–Fn3 (residues 349–737). Dissociation was measured in kinetics buffer for 600 s.

**Assays for survival, migration, and tube formation of ECs.** For survival assays, confluent HUVECs were incubated with 1 µg/ml of COMP-Angpt1, 10 µg/ml of hTAAB, or 10 µg/ml of hzTAAB-H2L2 in serum-free media for 36 h. Cells were photographed under a phase-contrast microscope and analyzed using a TdT-mediated dUTP-X nick end labeling (TUNEL) assay kit (Roche). Migration assays were performed using a modified Boyden chamber with polycarbonate filters (8 µm pores, Corning) coated with 0.1% gelatin. In total, $1 \times 10^5$ cells of HUVECs in 200 µl of serum-free media were seeded in the upper chamber of 12-well plates, and 600 µl of culture media was added into the lower chamber. The 1 µg/ml of COMP-Angpt1, 10 µg/ml of hTAAB, or 10 µg/ml of hzTAAB-H2L2 in serum-free media was added to the bottom chamber. After a 12-h incubation, non-migrated cells on the upper side of the filter were removed with a cotton swab. Migrated cells on the bottom surface of the filter were fixed with 4% formaldehyde, stained with crystal violet (Sigma-Aldrich), and counted at ×400 magnification under a microscope. For tube-formation assay, $4 \times 10^4$ cells of HUVECs were seeded on Matrigel (BD Biosciences)-coated 48-well plates and incubated with 1 µg/ml of COMP-Angpt1, 10 µg/ml of hTAAB or 10 µg/ml of hzTAAB-H2L2 in serum-free media for 16 h. After incubation, cells were photographed, and tube-forming activity was quantified by measuring the lengths of tube-like structures.

**Spheroid-based sprouting assay.** HUVEC spheroids were generated by culturing HUVECs at passage 4 in a culture medium containing 0.25% methylcellulose and incubating overnight as hanging drops. The spheroids were collected and embedded in 2 mg/ml collagen type I (Corning) and then treated with COMP-Angpt1 (1 µg/ml), hTAAB (10 µg/ml), or hzTAAB (10 µg/ml) for 48 h. The spheroids were fixed in 4% paraformaldehyde for 15 min at 4 °C and stained with anti-human CD31 (1:400 dilution, AF806, R&D Systems), and then imaged using a Cell observer (Carl Zeiss).

**Quantification and statistical analysis.** Data in figures and tables are reported as mean ± standard deviation (SD) or standard error of the mean (SEM) with the number of biological and technical replicates indicated in the figure and table legends where "n" represents the number of biological replicates performed. Immunoblotting, survival assay, migration assay, and tube-formation assay were assessed using the software ImageJ (v1.52) and GraphPad Prism (v8.0.2).

## Data availability
Coordinate file and structure factor file for the crystal structure of hTie2 Fn2–3/chimeric hTAAB Fab complex have been deposited in the Protein Data Bank (PDB) under accession code 7E72. All the other data and materials used for the analysis are available from the corresponding author upon reasonable request. Source data are provided with this paper.

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

## Acknowledgements

We are grateful to Prof. Yong-Seok Heo for providing modified pBAD expression vector and *E.coli* Top10F cells, and to the staff of beamline 11C at Pohang Accelerator Laboratory for help with the X-ray diffraction experiments and data collection. This work was supported by grants from the Institute for Basic Science (IBS-R025-D1 to G.Y.K. and IBS-R030-C1 to H.M.K.).

## Author contributions

G.J., J.B., G.Y.K., and H.M.K. designed the experiments and analyzed the data; G.J., H.J.H., A.-R.H., and J.A.K. purified the proteins and determined the crystal structure; G.J. performed the negative-stain EM analysis; J.B. generated Tie2-activating monoclonal antibody and performed the functional analysis of Tie2-activating antibody; D.-K.K., S.P.H., and S.L. performed cell biological experiments; G.J., J.B., G.Y.K., and H.M.K. wrote the manuscript.

## Competing interests

The authors declare no competing interests.
