## [Peer Review File · Nature Communications]

Structural insights into the clustering and activation of Tie2 receptor mediated by Tie2 agonistic antibody

Editorial Note: Parts of this Peer Review File have been redacted as indicated to remove third party material where no permission to publish were obtainedREVIEWER COMMENTS

Reviewer #1 (Remarks to the Author):

NCOMMS-21-14009-T

Review

This is very interesting and thorough manuscript on the design and humanization of an antibody that activates the Tie2 receptor through the membrane proximal fibronectin repeats. The authors conduct an impressive amount of work toward characterizing the mechanism of activation and the humanization of the antibody. However, the authors make several overly broad statements about the utility and novelty of the antibody with respects to earlier work. For example, I disagree with the novelty implied by the statement that “our findings provide insights into why higher-order oligomeric status of Angpt1 is critically required for Tie2 receptor clustering and activation”. This work primarily builds on several other crystal and small angle x-ray scattering structures including that of Ang/Tie2 complex and two independent Tie2 fibronectin repeat structures, and so it mainly confirms what was previously known and modeled as far back as the late 1990’s through rotary shadowing electron micrographs.

While the Tie2/Ab structure that was identified is no doubt interesting, it is no more a legitimate representation of Tie2 clustering than all other artificial models presented thus far. Similarly, the authors states that “this architecture contrasts with lateral Tie2 arrays previously observed in the crystal lattices”. This would suggest that others represented crystal contacts in the prior x-ray models as being legitimate representations of Tie2 clustering, which I don’t believe is truly the case. Thus, the major finding in this research that has strong implications for enhancing our understanding of Tie2 clustering is subject to the same critiques of all earlier works and therefore doesn’t rise to the level of providing strong new evidence.

Furthermore, the abstract states that “Angpt1 variants involve difficulties in production and storage”, while this is true it’s hard to envision how an antibody is going to be more effective at properly regulating Tie2 activation in vivo given that Angiopoietins have roles outside of Tie2 activation (fibronectin and integrin association, regulating Tie1/Tie2 heterodimerization, to name a few that we know of). It is unclear to me what the exact benefit to using an antibody is over the purified ligand since the ligand is not limited by potential immunogenic properties nor is the antibody able to influence the other multiple roles of either the ligand or the receptor. This suggests that this antibody might be best served as a clinic reagent (this remains to be seen) and not necessarily a tool to better understand Tie2 signaling. However, again, there are several other molecular tools already available that achieve the same goal with the same level of efficacy (with regards to Tie2 activation, one of them previously designed by the corresponding author).

In summary, this is an impressive amount of work, but the main result lies in the structural implications of receptor clustering, which are fairly small given what was already known and the limitations of the artificial nature of Tie2 clustering. It is true the authors generated a molecular tool to activate the Tie2 receptor, but several already exist and the authors have yet to demonstrate (outside of cell culture) that the antibody will have any benefit over the natural ligand (Ang1) particularly given the complex nature of the receptor and their interacting partners.

Minor points:

Figure 4c illustrates that the ectodomain is monomeric at what appears to be submicromolar concentrations. Yet AUC data from previous studies illustrates that it is dimeric at 2 μ M. Similarly, crystal structures including this one illustrates that Tie2 homodimerizes through the same interface previously identified by two independent labs at concentrations consistent with that found at the cell membrane.

Figure 1a – not sure if this is necessary. Most readers should understand how monoclonal antibodies are produced and there's nothing too novel about the approach used here.

Reviewer #2 (Remarks to the Author):

Employing a systematic hypothesis- and structure-driven experimental approach, the authors generated monoclonal mouse-anti-human anti-Tie2 antibodies that induce in a ligand-independent manner Tie2 clustering and subsequent activation. Using the most potent Tie2 activating antibodies, the authors

performed an in depth structure-function analysis to make the surprising and biologically important observation that Tie2 exists in its native, non-activated form as homodimers, which serves as starting point for multimerization upon ligand or antibody binding. Lastly, the authors humanize some of the antibodies and eliminate some of the activity losses of the AB resulting from the humanization process. Taken together, the authors propose to have generated a novel reagent for potential therapeutic application.

In conclusion, this is a technically highly ambitious and flawless study that can hardly be criticized. It would have been useful had the authors shown potential therapeutic use of the characterized antibody beyond highly reductionist cellular experiments. Yet, this reviewer acknowledges the difficulty of such approach, given that the AB is not cross-reacting with mouse Tie2, which would either require primate experimentation or experimentation with Tie2 humanized mice, which are not yet readily available. Therefore, as presented, the study remains somewhat speculative. Other means of Tie2 activation may have technical or biological limitations. Yet, whereas ABTAA would have a very unique therapeutic window, since it builds on the endogenous regulation of Ang2 production and converts antagonistic Ang2 into agonistic Ang2, the here presented Tie2 activating AB would promiscuously and systemically activate Tie2 in all Tie2 expressing vascular beds. Yet, this may not necessarily be the goal of therapeutic activation. In turn, sites where Tie2 activation may be beneficial, are likely also sites with Tie2 negative transcriptional regulation. Still, this conceptual concern does not take away from an otherwise very well designed and executed Tie2 structure-function study that will be an important addition to the literature.

Reviewer #3 (Remarks to the Author):

This manuscript reports development of a Tie2 agonist antibody that induces receptor activation by clustering pre-formed Tie2 dimers. Structural analysis reveals the mechanism of antibody binding to Fn3. Importantly, using disulphide-linked dimeric Tie2 ectodomain fragments and negative staining EM, the authors show the antibody clusters ectodomains into polygonal configurations. The agonist antibody is humanised, and optimised by some nice structure-based protein engineering. This antibody has significant potential for therapeutic development.

This work is of high quality, the data are convincing and the conclusions are clear and significant. Targeting the Tie2 Fn domains for antibody-induced clustering and receptor activation is novel. The finding that the antibody organises Tie2 into polygonal clusters is particularly interesting and raises the possibility that such clusters could also be involved in ligand-activation of Tie2 in cells.

Main points

1. The concept that ligand activation of Tie2 involves polygonal organisation of Tie2 ectodomain is suggested in the manuscript. If this was shown to be the case it would be a major advance in our understanding of Tie2 activation. Can the authors provide any data to support this model, for example, using COMP-Ang1 or native Ang1 and negative staining of ectodomain?

2. The focus of the manuscript is clustering and its association with receptor activation. The authors state that Angpt oligomeric status determines ability to cluster and thereby activate Tie2. However, others have shown that Ang2-FLD-Ang1 chimeras, as well as oligomeric Ang2 constructs bind but do not activate Tie2 (Procopio 1999, J Biol Chem 274, 30196; Nat Struct Biol 10, 38), indicating FLD is important in activation. Can the authors discuss this in the context of their clustering:activation model.

3. The data on antibody-induced clustering shown in Fig. 5b are from one or two cells for each treatment, and there is some indication of clustering in monomeric Tie2 0min. Conclusions from these data would be strengthened by showing more cells for each condition and possibly some form of quantification.

4. How do the large (μm) clusters of Tie2 induced by Ang1 and antibody (Fig 5) relate to the nm structures observed with ectodomain fragments in solution (Fig. 4)?

5. The data in Fig 6A is suggestive of a link between hinge flexibility of IgG and agonist activity but could be stronger - IgG2 and IgG4 show similar levels of Akt activation, are the differences between IgG's statistically significant? This conclusion needs modification.

Nicholas Brindle

Authors' response to the reviewers' comments
**“Structural insights into the clustering and activation of Tie2
receptor” by Jo and Bae et al.**

We are grateful to the reviewers for their careful evaluation of our manuscript and for constructive suggestions. In addition to revising the text, we have performed a series of experiments to address the reviewers' concerns. These new results have been incorporated into the revised manuscript. In the text below, the reviewers' comments are in italics and our responses and descriptions of the changes made in the manuscript are in bold.

Reviewer #1 (Remarks to the Author):

This is very interesting and thorough manuscript on the design and humanization of an antibody that activates the Tie2 receptor through the membrane proximal fibronectin repeats. The authors conduct an impressive amount of work toward characterizing the mechanism of activation and the humanization of the antibody. However, the authors make several overly broad statements about the utility and novelty of the antibody with respects to earlier work. For example, I disagree with the novelty implied by the statement that “our findings provide insights into why higher-order oligomeric status of Angpt1 is critically required for Tie2 receptor clustering and activation”. This work primarily builds on several other crystal and small angle x-ray scattering structures including that of Ang/Tie2 complex and two independent Tie2 fibronectin repeat structures, and so it mainly confirms what was previously known and modeled as far back as the late 1990's through rotary shadowing electron micrographs.

While the Tie2/Ab structure that was identified is no doubt interesting, it is no more a legitimate representation of Tie2 clustering than all other artificial models presented thus far. Similarly, the authors states that “this architecture contrasts with lateral Tie2 arrays previously observed in the crystal lattices”. This would suggest that others represented crystal contacts in the prior x-ray models as being legitimate representations of Tie2 clustering, which I don't believe is truly the case. Thus, the major finding in this research that has strong implications for enhancing our understanding of Tie2 clustering is subject to the same critiques of all earlier works and therefore doesn't rise to the level of providing strong new evidence.

We appreciate the reviewer's overall positive evaluation of our work. As the reviewer mentioned, it is well known that higher-order oligomeric status of Angpt1 is critically required for Tie2 receptor clustering and activation. Earlier studies with rotary shadowing transmission electron microscopy clearly visualized the multimeric nature of Tie2 agonists (native Angpt1 and COMP-Angpt1) [PMID: 12469114 and 15060279] and the structural features of the Tie2 extracellular domain, which consists of a globular head domain and a short rod-like stalk [PMID: 16849318]. In addition, crystal structures of the Angpt FLD/Tie2 Ig1-Ig3 complex [PMID: 16732286 and 23592718] and Tie2 fibronectin repeat [PMID:

28396439 and 28396397] provide insight into Angpt FLD-Tie2 receptor interaction and Tie2 dimerization. However, the precise structural features of the Tie2 cluster induced by Tie2 agonists (Native Angpt1, engineered Angpt variants, Tie2-agonistic antibody) were unclear.

The major finding in our study is that the binding of new Tie2 agonistic antibody (hTAAB) to membrane proximal Fn3 domains in Tie2 homodimer induces Tie2 polygonal clustering and activation. Moreover, our negative stain EM analysis is the first direct visualization of Tie2 clustering induced by Tie2 agonist. Interestingly, we also observed few linear Tie2 clusters from the Tie2 dimeric mutant/hTAAB IgG1 complex (Supplementary Fig. 4a, red box). Therefore, to investigate whether the different Tie2-agonistic activities of the IgG subclass is related to the Tie2 clustering mode (polygonal vs. linear), we performed shape-based statistical analysis of the heterogeneous Tie2 Ig3-Fn3 dimer/hTAAB IgG (human chimeric IgG1, IgG2, or IgG4) complex using negative stain EM. We classified individual particles into six categories according to their shape (linear, <tetragonal, tetragonal, pentagonal, hexagonal, and >hexagonal) and compared the Tie2 activation potency among IgG subclasses (Fig. 6c and Supplementary Fig. 5b and 5c).

Fig. 6 | c, Quantification of the heterogeneous composition of Tie2 dimeric mutant/hTAAB IgG subtype complex. The hTAAB IgGs (hlgG1, hlgG2, or hlgG4) were incubated with purified hTie2 Ig3-Fn3 N691C/D682C, followed by Size exclusion chromatography for Negative-stain EM analysis. Based on the shape of complex in negative-stain EM image, we classified particles into 6 categories (linear, <tetragonal, tetragonal, pentagonal, hexagonal, and >hexagonal form) and the proportion of each categories were indicated (Total counted particles n=1381, 1162, and 1277 for IgG1, IgG2, and IgG4, respectively).

Supplementary Fig. 5 | b, Size-exclusion chromatography of hTie2 Ig3-Fn3 D682C/N691C in complex with hTAAB IgGs (hIgG1, hIgG2, or hIgG4). Purified hTie2 Ig3-Fn3 dimeric mutant was incubated with hTAAB IgG1, IgG2, or IgG4 for 1 hr at a molar ratio of 2:1 (Tie2 monomer:hTAAB IgGs) and applied to size exclusion chromatography. The fraction used for negative-stain EM is indicated as EM analysis. **c**, Representative micrograph of negative-stain EM analysis of Tie2 Ig3-Fn3 dimeric mutant in complex with hTAAB IgGs (hIgG1, hIgG2, or hIgG4). Scale bars, 100 nm. The hTAAB IgGs-Tie2 dimer complex particles are classified into 6 categories (linear, <tetragonal, tetragonal, pentagonal, hexagonal, and >hexagonal form) according to their shape for quantification, and the representative particles are marked with a straight line for linear, triangle for <tetragonal, square for tetragonal, pentagon for pentagonal, hexagon for hexagonal, or circle for >hexagonal particles.

Notably, we found that the IgG2 subclass with the lowest Tie2-agonistic activity mostly induced the linear shape of Tie2 clustering, whereas various types of polygonal assembly were observed in the IgG1 and IgG4 subclasses, which exhibited higher Tie2-agonistic activity than IgG2 (tetragonal + pentagonal + hexagonal + >hexagonal clustering: 73.1%, 11.3%, and 69.5% for IgG1, IgG2, and IgG4, respectively).

The proportion of the linear form correlated with the hinge rigidity of the IgG subclass (IgG2>IgG4>IgG1) and inversely correlated with Tie2 and Akt phosphorylation (IgG1> IgG4>IgG2). These results indicate

that polygonal Tie2 clustering induced by hTAAB is critical for Tie2 activation. We have added these data to the revised manuscript (p. 12-13, highlighted in blue and Fig. 6c and Supplementary Fig. 5b and 5c).

As shown below, we also noted that this linear Tie2 clustering induced by hTAAB IgG2 (with the lowest Tie2-agonistic activity) was different from the lateral clustering proposed by Leppanen et al. based on a small angle X-ray scattering (SAXS) experiment with Angpt/Tie2 complex [PMID: 30036484]. [REDACTED]

Although the polygonal Tie2 clustering architecture induced by hTAAB IgG1 contrasts with the lateral Tie2 arrays that we observed in our crystal lattices (initially mentioned in the Discussion section), we would like to emphasize that we do not currently have a clear understanding of the Tie2 clustering mode induced by native Angpt1 on the endothelial cell membrane because native Angpt1 has a very heterogeneous and irregular oligomeric form, and it is even more difficult to visualize the structural features of the Tie2 cluster on the endothelial cell membrane.

We hope the reviewer understands the limitations of the methodologies we employed and agrees with our additional experiments supporting the significance of polygonal Tie2 clustering activation induced by hTAAB for Tie2 activation. We acknowledge that the polygonal Tie2 clustering model in the cellular membrane should be more rigorously examined in the future using a variety of approaches.

Furthermore, the abstract states that “Angpt1 variants involve difficulties in production and storage”, while this is true, it’s hard to envision how an antibody is going to be more effective at properly regulating Tie2 activation in vivo given that Angiopoietins have roles outside of Tie2 activation (fibronectin and integrin association, regulating Tie1/Tie2 heterodimerization, to name a few that we know of). It is unclear to me what the exact benefit to using an antibody is over the purified ligand since the ligand is not limited by potential immunogenic properties nor is the antibody able to influence the other multiple roles of either the ligand or the receptor. This suggests that this antibody might be best served as a clinic reagent (this remains to be seen) and not necessarily a tool to better understand Tie2 signaling. However, again, there are several other molecular tools already available that achieve the same goal with the same level of efficacy (with regards to Tie2 activation, one of them previously designed by the corresponding author).

We agree that our Tie2 agonistic antibody may not necessarily be a tool for better understanding Tie2 signaling because Tie2 cellular signaling *in vivo* can be further modulated by Tie1/Tie2 homodimer, vascular endothelial protein tyrosine phosphatase (VE-PTP), and integrins ($\alpha 5\beta 1$, $\alpha v\beta 3$, and $\alpha v\beta 5$). Moreover, unlike native Angpt1 and COMP-Angpt1, our antibody binds Tie2 at the membrane proximal FN domain, leading to only *cis*-clustering of Tie2 homodimers. These issues are initially mentioned in the Discussion section.

Circulating Angpt2 is markedly increased in a range of pathologies, including sepsis, diabetic retinopathy, and stroke, where it can antagonize Angpt1, leading to vessel regression, leakage, and inflammation. Therefore, the clinical development of drugs that target the ANG–TIE pathway has focused on Angpt2-blocking therapies (e.g., nesvacumab, Medi3617, LY3127804, trebanianib, vanucizumab).

Preclinical studies using models of stroke, sepsis, endotoxemia, and cardiac allograft vasculopathy have shown that administration of engineered Angpt1 variants can also prevent endothelial death and capillary loss and suppress vascular leakage and inflammation by activating Tie2 signaling. In contrast to Angpt2-blocking therapies, engineered Angpt1 variants have not yet been investigated in clinical trials, probably due to its poor production yield and manufacturability, which need to be addressed in the early stages of drug discovery. Therefore, we investigated the production yield, purification purity, and thermostability of COMP-Angpt1 and hzTAAB-H2L2 and compared their manufacturability as a therapeutic drug (Fig. 1 for reviewer only).

Fig. 1 for reviewer only | a, SDS-PAGE analysis of COMP-Angpt1 and hzTAAB-H2L2 after purification with affinity chromatography. **b**, Size exclusion chromatography of COMP-Angpt1 and hzTAAB-H2L2 after purification with affinity chromatography. Elution profiles were normalized to facilitate their comparison. **c**, Thermostability of COMP-Angpt1 and hzTAAB-H2L2, was determined by the thermofluor assay with SYPRO orange dye. The fluorescence signal is plotted as a function of temperature to get a sigmoidal melting curve of the proteins.

We obtained approximately 150 μg of COMP-Angpt1 and 1.5 mg of hzTAAB-H2L2 from affinity chromatography with 30 ml of suspension culture media (transient expression in Expi HEK293 cells). SDS-PAGE analysis of purified proteins from affinity chromatography and subsequent size-exclusion chromatography indicated that hzTAAB-H2L2 was uniform, whereas COMP-Angpt1 had a heterogeneous oligomeric status (Fig. 1a and b for reviewer only). Furthermore, thermostability analyses of COMP-Angpt1 and hzTAAB-H2L2 under conditions of long incubation (3 days) at room temperature or 4 $^{\circ}\text{C}$ clearly showed that hzTAAB-H2L2 is much more stable than COMP-Angpt1 during storage (Fig. 1c for reviewer only). Thus, hzTAAB-H2L2 has superior production yield and manufacturability compared to COMP-Angpt1. We present these data here only for the reviewer's consideration.

More importantly, to overcome antagonism by Angpt2 in a pathological condition, the level of exogenous Angpt1 variants should be high enough, but this raises the possibility of deleterious effects,

such as increased vessel remodeling in established vessels. Therefore, we recently developed a humanized monoclonal anti-Angpt2 antibody, Ang2-binding and Tie2-activating antibody (ABTAA), which can convert antagonistic Angpt2 to Tie2 agonist. However, ABTAA functions only in the presence of Angpt2, so its therapeutic efficacy largely depends on Angpt2 levels in the pathological microenvironment, which are difficult to control. Although ABTAA exhibited therapeutic potential in the NV-AMD and sepsis mouse model, the epitopes in Angpt2 for ABTAA binding and the oligomerization status of active signaling complexes (ABTAA/Angpt2/Tie2 complex) remain unclear. Moreover, the optimal dose of ABTAA for reducing unanticipated aggregation between ABTAA and Angpt2 needs to be carefully explored for clinical application.

Our in-depth structure-function analysis of new Tie2-activating antibody (hTAAB) revealed a clear mode of action of hTAAB for Tie2 activation, which can be a benefit in future clinical investigations, such as dose optimization and the prediction of clinical outcomes and side effects. Furthermore, our new experiments showed that, even in the presence Angpt2, which suppresses Angpt1 binding and Angpt1's protective effects, hTAAB binds at a site on Tie2 distinct from the Angpt-binding site, making it resistant to antagonism by Angpt2, similar to Tie2-synthetic ligand-1 (TSL1), which is not subject to antagonism or modulation by endogenous Angpt2 [PMID: 30036484]. Therefore, hTAAB that is not inhibited by Ang2 could potentially be used at lower concentrations, decreasing the likelihood of side effects. However, the *in vivo* biological effects of our hTAAB and hzTAAB under pathological conditions should be compared to ABTAA and other Tie2 agonists using appropriate animal models (see below). We have added these data to the revised manuscript (p. 15, highlighted in blue and Fig. 7g).

Fig. 7 | g, Immunoblot analysis for relative phosphorylation ratios of Tie2 and Akt after treatment of HUVECs with ABTAA or hzTAAB-H2L2 (10 µg/ml) with or without Angpt2 (1 µg/ml) (means ± SD, n = 3; ***p < 0.001 vs. control; ##p < 0.01 ABTAA vs. Angpt2+ABTAA; ###p < 0.001 ABTAA vs. Angpt2+ABTAA).

In summary, this is an impressive amount of work, but the main result lies in the structural implications of receptor clustering, which are fairly small given what was already known and the limitations of the artificial nature of Tie2 clustering. It is true the authors generated a molecular tool to activate the Tie2 receptor, but several already exist and the authors have yet to demonstrate (outside of cell culture) that the antibody will have any benefit over the natural ligand (Ang1) particularly given the complex nature of the receptor and their interacting partners.

Because hTAAB does not bind to mouse Tie2, we admitted that this study has a limitation to verify the biological and therapeutic effects of hTAAB in the mouse. We are currently generating humanized Tie2 mice but it will take a year to get them (see below, Fig. 2 for reviewer only). Nevertheless, to expect comparable, possible benefit of hTAAB and hzTAAB-H2L2 over the COMP-Ang1 *in vivo*, we additionally performed a HUVEC 3D spheroid assay. As a result, all COMP-Ang1, hTAAB and hzTAAB-H2L2 similarly not only increased sprouting *per se* but also induced enlarged sprouting in the absence and presence of VEGF-A. Vascular enlargement is one of hallmarks of Ang1-induced biological effects *in vivo*. Given that the enlarged sprouting *in vitro* could reflect the vascular enlargement *in vivo*, both hTAAB and hzTAAB-H2L2 could have comparable benefit *in vivo* through Tie2 activation although native Ang1 could have another possible beneficial effect through interacting non-Tie2 partners. We have added these data to the revised manuscript (p. 15, highlighted in blue and Fig. 7f).

Fig. 7 | f, Representative images of HUVEC spheroids 3D culture after stimulation with COMP-Angpt1 (1 $\mu\text{g}/\text{ml}$), hTAAB (10 $\mu\text{g}/\text{ml}$), or hzTAAB (10 $\mu\text{g}/\text{ml}$) with or without VEGF-A for 48 hr (left). Scale bars, 100 μm . Comparison of the number and width of sprouts is shown in right panel and each dot indicates a value from $n = 4\sim 6$ spheroids/group (mean \pm SD, ** $p < 0.01$ vs. control).

According to our structural information for the hTie2 Fn2–3/chimeric hTAAB Fab complex and the sequence comparison between human and mouse Tie2, we predicted that a long charged side chain

(R729) in mouse and rat Tie2 that corresponds to the hydrophobic valine (V730) of human Tie2 at the center of the hTAAB binding interface is the major cause of inhibition of hTAAB binding. To confirm this, we mutated the R729 residue of mouse Tie2 to valine and analyzed its binding to hTAAB H2L2 and its phosphorylation in HEK cells after treatment with hTAAB H2L2.

Fig. 2 for reviewer only | a, Pull-down assay of mouse Tie2 R729V mutant with hTAAB-H2L2. The purified mouse Tie2 Ig3-Fn3 R729V mutant was pull-down with the hTAAB-H2L2-bounded protein A resin. After washing with PBS, resin (R) and flow-through (FT) were subjected to SDS-PAGE analysis. The human and mouse Tie2 Ig3-Fn3 WT were used as a positive and negative control, respectively. The bound human or mouse Tie2 Ig3-Fn3 is indicated by the red box. **b**, Binding kinetics of hTAAB-H2L2 for mouse Tie2 Ig3-Fn3 R729V mutant determined by SPR. **c**, Relative phosphorylation ratios of Tie2 were analyzed by immunoblotting with HEK293T cells transiently expressing full-length human Tie2 WT, and mouse Tie2 WT or R729V mutant after hTAAB-H2L2 (10 $\mu\text{g/ml}$) treatment for 1 hr.

As expected, the single residue mutation (R729V in mouse Tie2) enabled hTAAB to bind mouse Tie2 and induce Tie2 phosphorylation. Based on these observations, we have started to generate Tie2 humanized mice (Tie2 R729V mutant mouse) for further investigation of the therapeutic efficacy of hTAAB in the *in vivo* disease model. It will also be of interest in future studies to examine the effects of hTAAB in pathological conditions in which Angpt2 is elevated and to compare the effect of hTAAB to that of other Tie2 agonists (COMP-Angpt1, ABTAA, etc.) using Tie2 humanized mice. We hope the reviewer understands and agrees that analyzing the therapeutic effect of hTAAB with Tie2 humanized mice or primate is beyond the scope of our current study. However, we do hope to address these issues in a future follow-up study.

Minor points:

Figure 4c illustrates that the ectodomain is monomeric at what appears to be submicromolar concentrations. Yet AUC data from previous studies illustrates that it is dimeric at 2 μ M. Similarly, crystal structures including this one illustrates that Tie2 homodimerizes through the same interface previously identified by two independent labs at concentrations consistent with that found at the cell membrane.

The concentrations of Tie2 Fn2-3, Tie2 ECD, hTAAB/Tie2 Fn2-3 complex, and hTAAB/Tie2 ECD complex that we used in the SEC-MALS experiment (Fig. 4c) were 3 mg/ml (134.32 μ M), 1 mg/ml (12.42 μ M), 1 mg/ml (5.8 μ M), and 1 mg/ml (4.34 μ M), respectively. Our results are consistent with the SEC-MALS results in the previous study [PMID: 28396439] in which they used even higher concentrations (3 or 6 mg/ml) of Tie2 ECD and Fn1-3. However, their data from small-angle X-ray scattering (SAXS) using an Fn2-3 protein (~80 μ M) indicated a structure that fits best with their crystal structure of an Fn3 domain-mediated Tie2 dimer. Sedimentation equilibrium analytical ultracentrifugation (SE-AUC) in the accompanying paper showed that Tie2 ECD is a dimer at 2 μ M [PMID: 28396397], as the reviewer mentioned. These contrasting results suggest that the unliganded receptor has a relatively weak tendency for the Fn-domain-mediated interactions and that the putative homotypic interactions by the full-length receptor require spatial (and dimensional) constraints in the plasma membrane and/or is ligand-dependent.

To investigate whether WT Tie2 exists as a dimer in the cell membrane, we purified full-length Tie2-GFP WT and compared the size to that of constitutive dimeric Tie2 mutant using fluorescence size-exclusion chromatography (FSEC). FSEC profiles with detergent-solubilized full-length Tie2-GFP (WT and constitutive dimeric mutant) clearly demonstrated that Tie2 WT is a dimer in the plasma membrane. We have added these data to the revised manuscript (p. 11, highlighted in blue and Fig. 5d).

Fig. 5 | d, Fluorescence-size exclusion chromatography (FSEC) analysis of full-length Tie2-GFP WT, dimeric or monomeric Tie2 mutant, which are solubilized from HEK293F membrane by LMNG-CHS.

Figure 1a – not sure if this is necessary. Most readers should understand how monoclonal antibodies are produced and there's nothing too novel about the approach used here.

In response to the reviewer's comment, we have removed the schematic diagram for the monoclonal antibody generation strategy in Figure 1a but kept the schematic of the domain structure of Tie2 receptor in which we indicated the Tie2 domains used for mouse immunization.

Reviewer #2 (Remarks to the Author):

Employing a systematic hypothesis- and structure-driven experimental approach, the authors generated monoclonal mouse-anti-human anti-Tie2 antibodies that induce in a ligand-independent manner Tie2 clustering and subsequent activation. Using the most potent Tie2 activating antibodies, the authors performed an in depth structure-function analysis to make the surprising and biologically important observation that Tie2 exists in its native, non-activated form as homodimers, which serves as starting point for multimerization upon ligand or antibody binding. Lastly, the authors humanize some of the antibodies and eliminate some of the activity losses of the AB resulting from the humanization process. Taken together, the authors propose to have generated a novel reagent for potential therapeutic application.

In conclusion, this is a technically highly ambitious and flawless study that can hardly be criticized. It would have been useful had the authors shown potential therapeutic use of the characterized antibody beyond highly reductionist cellular experiments. Yet, this reviewer acknowledges the difficulty of such approach, given that the AB is not cross-reacting with mouse Tie2, which would either require primate experimentation or experimentation with Tie2 humanized mice, which are not yet readily available. Therefore, as presented, the study remains somewhat speculative.

We appreciate the reviewer's assessment of our work as 'highly ambitious and flawless study that can hardly be criticized'.

In response to the reviewer's comment, we have performed additional experiments.

Because hTAAB does not bind to mouse Tie2, we admitted that this study has a limitation to verify the biological and therapeutic effects of hTAAB in the mouse. We are currently generating humanized Tie2 mice but it will take a year to get them (see below, Fig. 1 for reviewer only). Nevertheless, to expect comparable, possible benefit of hTAAB and hzTAAB-H2L2 over the COMP-Ang1 *in vivo*, we additionally performed a HUVEC 3D spheroid assay. As a result, all COMP-Ang1, hTAAB and hzTAAB-H2L2 similarly not only increased sprouting *per se* but also induced enlarged sprouting in the absence and presence of VEGF-A. Vascular enlargement is one of hallmarks of Ang1-induced biological effects *in vivo*. Given that the enlarged sprouting *in vitro* could reflect the vascular enlargement *in vivo*, both hTAAB and hzTAAB-H2L2 could have comparable benefit *in vivo* through Tie2 activation although native Ang1 could have another possible beneficial effect through interacting non-Tie2 partners. We have added these data to the revised manuscript (p. 15, highlighted in blue and Fig. 7f).

Fig. 7 | f, Representative images of HUVEC spheroids 3D culture after stimulation with COMP-Angpt1 (1 $\mu\text{g}/\text{ml}$), hTAAB (10 $\mu\text{g}/\text{ml}$), or hzTAAB (10 $\mu\text{g}/\text{ml}$) with or without VEGF-A for 48 hr (left). Scale bars, 100 μm . Comparison of the number and width of sprouts is shown in right panel and each dot indicates a value from $n = 4\sim 6$ spheroids/group (mean \pm SD, ** $p < 0.01$ vs. control).

According to our structural information for the hTie2 Fn2–3/chimeric hTAAB Fab complex and the sequence comparison between human and mouse Tie2, we predicted that a long charged side chain (R729) in mouse and rat Tie2 that corresponds to the hydrophobic valine (V730) of human Tie2 at the center of the hTAAB binding interface is the major cause for inhibition of hTAAB binding. To confirm this, we mutated the R729 residue of mouse Tie2 to valine and analyzed its binding to hzTAAB H2L2 and its phosphorylation in HEK cells after treatment with hzTAAB H2L2.

Fig. 1 for reviewer only | a, Pull-down assay of mouse Tie2 R729V mutant with hzTAAB-H2L2. The purified mouse Tie2 Ig3-Fn3 R729V mutant was pull-down with the hzTAAB-H2L2-bounded protein A resin. After washing with PBS, resin (R) and flow-through (FT) were subjected to SDS-PAGE analysis. The human and mouse Tie2 Ig3-Fn3 WT were used as a positive and negative control, respectively. The bound human or mouse Tie2 Ig3-Fn3 is indicated by the red box. **b**, Binding kinetics of hzTAAB-H2L2 for mouse Tie2 Ig3-Fn3 R729V mutant determined by SPR. **c**, Relative phosphorylation ratios of Tie2 were analyzed by immunoblotting with HEK293T cells transiently expressing full-length human Tie2 WT, and mouse Tie2 WT or R729V mutant after hzTAAB-H2L2 (10 $\mu\text{g}/\text{ml}$) treatment for 1 hr.

As expected, the single residue mutation (R729V in mouse Tie2) enabled hzTAAB to bind the mouse Tie2 and induce Tie2 phosphorylation. Based on these observations, we have started to generate Tie2 humanized mice (Tie2 R729V mutant mouse) for further investigation of hzTAAB's therapeutic efficacy in the *in vivo* disease model. We hope the reviewer understands and agrees that analyzing the therapeutic effect of hzTAAB with Tie2 humanized mice or primate is beyond the scope of our current study. However, we do hope to address these issues in a future follow-up study.

Other means of Tie2 activation may have technical or biological limitations. Yet, whereas ABTAA would have a very unique therapeutic window, since it builds on the endogenous regulation of Ang2 production and converts antagonistic Ang2 into agonistic Ang2, the here presented Tie2 activating AB would promiscuously and systemically activate Tie2 in all Tie2 expressing vascular beds. Yet, this may not necessarily be the goal of therapeutic activation. In turn, sites where Tie2 activation may be beneficial, are likely also sites with Tie2 negative transcriptional regulation. Still, this conceptual concern does not take away from an otherwise very well designed and executed Tie2 structure-function study that will be an important addition to the literature.

This comment is related to comment #2 from Reviewer 1. Ang2 engages different signaling pathways in quiescent ECs versus angiogenic ECs. In resting blood vessels, Ang2 acts primarily as a vessel-destabilizing factor through Tie2 antagonism. As reviewer mentioned, during sprouting angiogenesis, tip ECs decrease their Tie2 expression, yet Ang2 can still regulate EC migration via integrin signaling. Because circulating Ang2 is markedly increased in a range of pathologies, including sepsis, diabetic retinopathy, and stroke, in which it can antagonize Ang1, leading to vessel regression, leakage, and inflammation, clinical development of drugs that target the ANG-TIE pathway has focused on Angpt2-blocking therapies (e.g., nesvacumab, Medi3617, LY3127804, trebanianib, vanucizumab).

Preclinical studies of engineered Angpt1 using models of stroke, sepsis, endotoxemia, and cardiac allograft vasculopathy have shown that exogenous Angpt1 can also prevent endothelial death and capillary loss and suppress vascular leakage and inflammation by activating Tie2 signaling. In contrast to Angpt2-blocking therapies, engineered Angpt1 has not yet been investigated in clinical trials. More importantly, to overcome antagonism by Angpt2 in pathological conditions, the level of exogenous Angpt1 should be high enough, which increases the possibility of deleterious effects, such as increased vessel remodeling in established vessels.

As the reviewer mentioned, ABTAA converts antagonistic Ang2 into agonistic Ang2. However, it functions only in the presence of Angpt2, so its therapeutic efficacy largely depends on Angpt2 levels in a pathological microenvironment, which are difficult to control. Moreover, the epitopes in Angpt2 for ABTAA binding and the oligomerization status of active signaling complexes (ABTAA/Angpt2/Tie2 complex) remain unclear. Our new experiment clearly demonstrated that, even in the presence Angpt2, which suppresses Angpt1 binding and Angpt1's protective effects, hTAAB binds at a site on Tie2 distinct from the Angpt-binding site, making it resistant to antagonism by Ang2, similar to Tie2-synthetic ligand-1 (TSL1), which is not subject to antagonism or modulation by endogenous Angpt2 [PMID: 30036484]. Therefore, hTAAB that is not inhibited by Ang2 could potentially be used at lower concentrations, decreasing the likelihood of side effects. We have added these data to the revised manuscript (p. 15, highlighted in blue and Fig. 7g).

Fig. 7 | g, Immunoblot analysis for relative phosphorylation ratios of Tie2 and Akt after treatment of HUVECs with ABTAA or hzTAAB-H2L2 (10 μ g/ml) with or without Angpt2 (1 μ g/ml) (means \pm SD, n = 3; ***p < 0.001 vs. control; ##p < 0.01 ABTAA vs. Angpt2+ABTAA; ###p < 0.001 ABTAA vs. Angpt2+ABTAA).

Despite Angpt1 variants, ABTAA and our Tie2-activating antibody (hTAAB) having the same ultimate goal of activating Tie2, they function in slightly different ways, thus likely to lead to somewhat different therapeutic outcome under pathological condition. Therefore, it will be of interest in future studies to examine the therapeutic effects of hzTAAB under pathological conditions in which Angpt2 is elevated and to compare the effect of hzTAAB to that of other Tie2 agonists (COMP-Angpt1, ABTAA, etc.) using Tie2 humanized mice. Moreover, combined treatment of hzTAAB with Angpt2-blocking antibody can be also investigated.

Reviewer #3 (Remarks to the Author):

This manuscript reports development of a Tie2 agonist antibody that induces receptor activation by clustering pre-formed Tie2 dimers. Structural analysis reveals the mechanism of antibody binding to Fn3. Importantly, using disulphide-linked dimeric Tie2 ectodomain fragments and negative staining EM, the authors show the antibody clusters ectodomains into polygonal configurations. The agonist antibody is humanised, and optimised by some nice structure-based protein engineering. This antibody has significant potential for therapeutic development.

This work is of high quality, the data are convincing and the conclusions are clear and significant. Targeting the Tie2 Fn domains for antibody-induced clustering and receptor activation is novel. The finding that the antibody organises Tie2 into polygonal clusters is particularly interesting and raises the possibility that such clusters could also be involved in ligand-activation of Tie2 in cells.

We appreciate the reviewer's commendation of our study as "significant" and "novel". We have fully revised the manuscript to address the reviewer's major criticism and hope that the reviewer is satisfied with our additional data.

Main points

1. The concept that ligand activation of Tie2 involves polygonal organisation of Tie2 ectodomain is suggested in the manuscript. If this was shown to be the case it would be a major advance in our understanding of Tie2 activation. Can the authors provide any data to support this model, for example, using COMP-Ang1 or native Ang1 and negative staining of ectodomain?

We appreciate the reviewer's insightful suggestion. To address this, we investigated the structures of the hTie2 ECD dimeric mutant (D682C/N691C), COMP-Angpt1, and their complex by negative-stain electron microscopy. These experiments demonstrated that the hTie2 ECD dimeric mutant (D682C/N691C) has a V-shape structure as expected and COMP-Angpt1 has a heterogeneous oligomeric status, which is consistent with previous reports. We also observed an octopus-like radial cluster in the Tie2 ECD dimeric mutant (D682C/N691C)/COMP-Angpt1 mixture. These architectures are similar to the proposed model in Fig. S6D (left), suggesting that both hTAAB IgG1 and COMP-Angpt1 induce the polygonal Tie2 cluster in a similar manner, though how native Angpt1 induces the Tie2 cluster is not yet clear. However, the cluster of Tie2 ECD dimeric mutant (D682C/N691C)/COMP-Angpt1 complex is not uniform because of the heterogeneous oligomeric COMP-Angpt1, and it is difficult to define individual domains of the Tie2 ECD dimeric mutant (D682C/N691C) and COMP-Angpt1. Due to this ambiguity, we

have not included these data in the revised manuscript and present it here only for the reviewer's consideration.

Fig. 1 for reviewer only | a, Negative-stain EM of purified hTie2 ECD dimeric mutant (D682C/N691C). b, Negative-stain EM of COMP-Angpt1. c, Negative-stain EM of Tie2 ECD dimeric mutant and COMP-Angpt1 complex. Scale bars, 100 nm.

2. The focus of the manuscript is clustering and its association with receptor activation. The authors state that Angpt oligomeric status determines ability to cluster and thereby activate Tie2. However, others have shown that Ang2-FLD-Ang1 chimeras, as well as oligomeric Ang2 constructs bind but do not activate Tie2 (Procopio 1999, *J Biol Chem* 274, 30196; *Nat Struct Biol* 10, 38), indicating FLD is important in activation. Can the authors discuss this in the context of their clustering:activation model.

The earlier research that the reviewer mentioned showed that the distinguishing features of Angpt2 compared to Angpt1 can be found within its FLD domain [PMID: 10514510 and 12469114]. However, the crystal structures and biochemical analyses demonstrated that the FLDs of Angpt1 and Angpt2 bind to the same Ig2 domain of Tie2 with similar affinities [PMID: 16732286 and 23592718]. In addition, FLD binding does not cause any significant conformational re-arrangement in the Tie2 ECD, indicating that Tie2 clustering and activation are most likely specified by Angpt oligomeric status (initially mentioned

in the introduction section). However, we do not currently have a clear understanding of the Tie2 clustering mode induced by native Angpt1 on the endothelial cell membrane because native Angpt1 has a heterogeneous and irregular oligomeric form. Moreover, it is even more difficult to visualize the structural feature of the Tie2 cluster on the endothelial cell membrane and we do not fully understand the complex roles of Angpts outside of Tie2 activation, such as fibronectin and integrin association, regulation of Tie1/Tie2 heterodimerization, or the significance of *cis*- or *trans*-Tie2 clustering in endothelial cells, hematopoietic stem cells, and pericytes.

Our in-depth structure-function analysis of new Tie2-activating antibody (hTAAB) revealed a clear mode of action of hTAAB for Tie2 activation, which can be a benefit in future clinical investigations, such as dose optimization and the prediction of clinical outcomes and side effects. Furthermore, our new experiments showed that, even in the presence Angpt2, which suppresses Angpt1 binding and Angpt1's protective effects, hTAAB binds at a site on Tie2 distinct from the Angpt-binding site, making it resistant to antagonism by Angpt2, similar to Tie2-synthetic ligand-1 (TSL1), which is not subject to antagonism or modulation by endogenous Angpt2 [PMID: 30036484]. Therefore, hTAAB that is not inhibited by Ang2 could potentially be used at lower concentrations, decreasing the likelihood of side effects. However, the *in vivo* biological effects of our hTAAB and hzTAAB under pathological conditions should be compared to ABTAA and other Tie2 agonists using appropriate animal models (see below). We have added these data to the revised manuscript (p. 15, highlighted in blue and Fig. 7g).

Fig. 7 | g, Immunoblot analysis for relative phosphorylation ratios of Tie2 and Akt after treatment of HUVECs with ABTAA or hzTAAB-H2L2 (10 μ g/ml) with or without Angpt2 (1 μ g/ml) (means \pm SD, n = 3; ***p < 0.001 vs. control; ##p < 0.01 ABTAA vs. Angpt2+ABTAA; ###p < 0.001 ABTAA vs. Angpt2+ABTAA).

We hope the reviewer understands the limitations of the methodologies we employed and agrees with the significance of polygonal Tie2 clustering for Tie2 activation induced by Tie2 agonistic antibody

hTAAB. We acknowledge that this model should be more rigorously examined with native Angpt in the future using a variety of approaches.

3. The data on antibody-induced clustering shown in Fig. 5b are from one or two cells for each treatment, and there is some indication of clustering in monomeric Tie2 0min. Conclusions from these data would be strengthened by showing more cells for each condition and possibly some form of quantification.

We appreciate the reviewer's constructive suggestions. In Fig. 5b, we over-expressed the full-length Tie2-GFP (WT, dimeric, or monomeric mutant) in HEK293 cells, and it is difficult to control their expression level in each cell. Therefore, as reviewer mentioned, highly over-expressed full-length Tie2 in some cells can lead to clustering by themselves even without ligand treatment (COMP-Angpt1, hTAAB).

In response to the reviewer's comment, we have selected cells that do not contain any Tie2 cluster at 0 min and performed additional time-lapse imaging with ligand treatment (COMP-Angpt1, hTAAB). For quantification of Tie2 clustering, we defined a cluster as discrete GFP puncta with a fluorescence intensity of 50–255 a.u. and size > 0.2 μm EqDiameter and measured the total area for GFP puncta per cell in 10 different cells for each condition using the 'Objective Count' tool in the Nikon imaging software. These quantification analyses confirmed that the clusters of Tie2 WT and constitutive Tie2 dimers are triggered at the cell surface upon treatment with hTAAB or COMP-Angpt1, whereas constitutive Tie2 monomers failed to form clusters. These results indicate that ligand-independent Tie2 dimerization is required for hTAAB-mediated Tie2 clustering, similar to COMP-Angpt1. We have added these data to the revised manuscript (p. 11, highlighted in blue and Fig. 5c).

Fig. 5 | Quantification (c) of the total area of GFP puncta per cell after treatment with COMP-Angpt1 or hTAAB is shown (means \pm SEM, n = 10; *p < 0.05, **p < 0.01, ***p < 0.001 vs. control).

4. How do the large (μm) clusters of Tie2 induced by Ang1 and antibody (Fig 5) relate to the nm structures observed with ectodomain fragments in solution (Fig. 4)?

As the reviewer mentioned, there is a size gap between the μm -scale clusters observed on the cell membrane (Fig. 5b, fluorescence intensity of 50–255 a.u. and size $> 0.2 \mu\text{m}$ EqDiameter) and the nm-scale Tie2 polygonal cluster ($\sim 50 \text{ nm}$) observed in negative-stain EM (Fig. 4g-h). Large clusters of Tie2 on the cell membrane are likely to be induced by hTAAB and COMP-Angpt1 as follows. We present the model here for the reviewer's consideration because we need more data to support this model. However, we have mentioned the relevant results in the appropriate section of the revised text.

Fig. 2 for reviewer only | Schematic representation of hypothetical clustering complex of Tie2 induced by hTAAB or COMP-Angpt1 on cell membrane. As the binding of hTAAB and COMP-Angpt1 to Tie2 are randomly occurred on cellular membrane, the polygonal clustering of Tie2 induced by hTAAB and COMP-Angpt1 can be further extended to a larger cluster.

5. The data in Fig 6A is suggestive of a link between hinge flexibility of IgG and agonist activity but could be stronger - IgG2 and IgG4 show similar levels of Akt activation, are the differences between IgG's statistically significant? This conclusion needs modification.

We apologize for the inadequate quality of our original data. We re-performed and compared Tie2-agonistic activities between IgG subclasses (chimeric IgG1, IgG2, or IgG4, n=5) in HUVECs. Indeed, Tie2 and Akt phosphorylation was induced at significantly higher levels by IgG1 and IgG4 than IgG2 treatment.

Fig. 6 | a and b, Immunoblot analysis for relative phosphorylation ratios of Tie2 and Akt after treatment of HUVECs with hTAAB in the form of mouse IgG1 or human chimeric IgG1, IgG2 or IgG4 (1 μ g/ml) **(a)**. Densitometric analyses **(b)** of p-Tie2/Tie2 and p-Akt/Akt ratios are shown (means \pm SD, n = 5; *p < 0.05, **p < 0.01, ***p < 0.001 vs. control; ###p < 0.001 hlgG1 vs. hlgG2; \$\$p < 0.01 hlgG2 vs. hlgG4).

Our current research using negative stain EM directly visualized the polygonal clustering of the Tie2 extracellular domain upon binding of Tie2 agonistic antibodies (hTAAB). Interestingly, we also observed few linear Tie2 clusters from the Tie2 dimeric mutant/hTAAB IgG1 complex (Supplementary Fig. 4a, red box). Therefore, to investigate whether the different Tie2-agonistic activities of IgG subclasses are related to the Tie2 clustering mode (polygonal vs. linear), we performed shape-based statistical analysis of heterogeneous Tie2 Ig3-Fn3 dimer/hTAAB IgGs (human chimeric IgG1, IgG2, or IgG4) complex using negative stain EM. We classified individual particles into six categories according to their shape (linear, <tetragonal, tetragonal, pentagonal, hexagonal, and >hexagonal) and compared Tie2 activation potency among IgG subclasses.

Fig. 6 | c, Quantification of the heterogeneous composition of Tie2 dimeric mutant/hTAAB IgG subtype complex. The hTAAB IgGs (hlgG1, hlgG2, or hlgG4) were incubated with purified hTie2 Ig3-Fn3 N691C/D682C, followed by Size exclusion chromatography for Negative-stain EM analysis. Based on the shape of complex in negative-stain EM image, we classified particles into 6 categories (linear, <tetragonal, tetragonal, pentagonal, hexagonal, and >hexagonal form) and the proportion of each categories were indicated (Total counted particles n=1381, 1162, and 1277 for IgG1, IgG2, and IgG4, respectively).

Supplementary Fig. 5 | b, Size-exclusion chromatography of hTie2 Ig3-Fn3 D682C/N691C in complex with hTAAB IgGs (hIgG1, hIgG2, or hIgG4). Purified hTie2 Ig3-Fn3 dimeric mutant was incubated with hTAAB IgG1, IgG2, or IgG4 for 1 hr at a molar ratio of 2:1 (Tie2 monomer:hTAAB IgGs) and applied to size exclusion chromatography. The fraction used for negative-stain EM is indicated as EM analysis. **c**, Representative micrograph of negative-stain EM analysis of Tie2 Ig3-Fn3 dimeric mutant in complex with hTAAB IgGs (hIgG1, hIgG2, or hIgG4). Scale bars, 100 nm. The hTAAB IgGs-Tie2 dimer complex particles are classified into 6 categories (linear, <tetragonal, tetragonal, pentagonal, hexagonal, and >hexagonal form) according to their shape for quantification, and the representative particles are marked with a straight line for linear, triangle for <tetragonal, square for tetragonal, pentagon for pentagonal, hexagon for hexagonal, or circle for >hexagonal particles.

Notably, we found that the IgG2 subclass with the lowest Tie2-agonistic activity mostly induced linear Tie2 clustering, whereas various types of polygonal assembly were observed in the IgG1 and IgG4 subclasses, which exhibited higher Tie2-agonistic activity than IgG2 (tetragonal + pentagonal + hexagonal + >hexagonal clustering: 73.1%, 11.3%, and 69.5% for IgG1, IgG2, and IgG4, respectively). The proportion of the linear form correlated with the hinge rigidity of the IgG subclass (IgG2>IgG4>IgG1) and inversely correlated with Tie2 and Akt phosphorylation (IgG1> IgG4>IgG2). These results indicate that polygonal Tie2 clustering induced by hTAAB is critical for Tie2 activation, and this architecture contrasts with the lateral Tie2 arrays observed in our crystal lattices. We have added these data to the revised manuscript (p. 12~13, highlighted in blue and Fig. 6c and Supplementary Fig. 5b and 5c).

Again, we thank all of the reviewers for their valuable assessments of our manuscript. We have carefully revised the manuscript based on their comments and hope that the revised manuscript is now acceptable for publication in *Nature Communications*.

REVIEWERS' COMMENTS

Reviewer #2 (Remarks to the Author):

No further comments - congratulations to a well and thoroughly revised manuscript!

Reviewer #3 (Remarks to the Author):

The authors have addressed my concerns and provided additional valuable and interesting data.

RESPONSE TO THE REVIEWERS' COMMENTS

Reviewer #2 (Remarks to the Author):

No further comments - congratulations to a well and thoroughly revised manuscript!

We appreciate these favorable and encouraging comments.

Reviewer #3 (Remarks to the Author):

The authors have addressed my concerns and provided additional valuable and interesting data.

We thank the reviewer for acknowledging the quality of our revision.